# DIG-MILP: a Deep Instance Generator for Mixed-Integer Linear Programming with Feasibility Guarantee

## Abstract

Mixed-integer linear programming (MILP) stands as a notable NP-hard problem pivotal to numerous crucial industrial applications. The development of effective algorithms, the tuning of solvers, and the training of machine learning models for MILP resolution all hinge on access to extensive, diverse, and representative data. Yet compared to the abundant naturally occurring data in image and text realms, MILP is markedly data deficient, underscoring the vital role of synthetic MILP generation. We present DIG-MILP, a deep generative framework based on variational auto-encoder (VAE), adept at extracting deep-level structural features from highly limited MILP data and producing instances that closely mirror the target data. Notably, by leveraging the MILP duality, DIG-MILP guarantees a correct and complete generation space as well as ensures the boundedness and feasibility of the generated instances. Our empirical study highlights the novelty and quality of the instances generated by DIG-MILP through two distinct downstream tasks: (S1) Data sharing, where solver solution times correlate highly positive between original and DIG-MILP-generated instances, allowing data sharing for solver tuning without publishing the original data; (S2) Data Augmentation, wherein the DIG-MILP-generated instances bolster the generalization performance of machine learning models tasked with resolving MILP problems.

## 1 Introduction

Mixed integer linear programming (MILP) is a prominent problem central to operations research (OR) (Achterberg & Wunderling, 2013; Wolsey, 2020). It forms the basis for modeling numerous crucial industrial applications, including but not limited to supply chain management (Hugos, 2018), production scheduling (Branke et al., 2015), financial portfolio optimization (Mansini et al., 2015), and network design (Al-Falahy & Alani, 2017; Radosavovic et al., 2020). This article aims to answer the question: *How can one produce a series of high-quality MILP instances?* The motivation behind this inquiry is illustrated through the subsequent scenarios:

**(Scenario I).** In industry, clients from real-world business seek specialized companies to develop or fine-tune intricate solver systems (Cplex, 2009; Bestuzheva et al., 2021; Gurobi, 2023) for solving MILP problems. The empirical success of the systems heavily depends on well-tuned hyperparameters for the solvers, which demands ample and representative testing cases that accurately reflect the actual cases. However, real data is often scarce during the early stages of a business. In addition, clients are typically reluctant to publish data that might encompass some specific information (e.g., schedules or contract stipulations for flight arrangement (Richards & How, 2002; Roling et al., 2008), platform costs or audience data for ad placements (Rodríguez et al., 2016)). These scenarios intensify the emergent need for generating instances that closely mirror the target data.

**(Scenario II).** In academia, beyond the improvement of algorithms (Lawler & Wood, 1966; Gamrath et al., 2015) for solving MILP, recent efforts have explored the use of machine learning (ML), which bypasses the need for expert knowledge and instead leverages historical data to foster accelerated resolutions (Khalil et al., 2016; 2017; Nair et al., 2020). Notably, the efficacy of ML-driven approaches relies on high-quality, large-capacity, and representative training data (Lu et al., 2022).

Given the scarce availability of real-world datasets (Gleixner et al., 2021), the scenarios mentioned above underscore the motivation to synthetically generate novel instances that resemble the limited existing MILP data. To meet the requirements of both the industrial and academic sectors, the challenge in synthetic MILP generation lies in ensuring authenticity, representativeness, and diversity. "Authenticity" refers to the general expectation in business scenarios that MILP problems should be bounded and feasible, where, otherwise, the modeling and the corresponding real-world problem would turn out to be meaningless. "Representativeness" means that the generated data should closely mirror the original data in terms of the problem scale and modeling logic

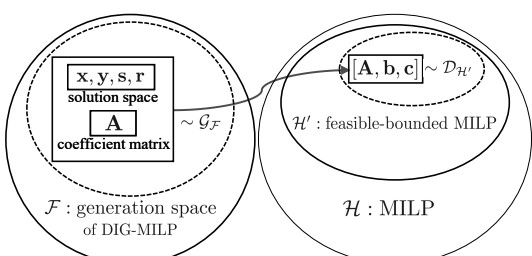

Figure 1: DIG-MILP generates feasible-bounded instances that resemble the target MILP data from distribution $\mathcal{D}_{\mathcal{H}'}$ by learning to sample the coefficient matrix along with a set of feasible solutions for both the primal format and dual format of the linear relaxation from the corresponding distribution $\mathcal{G}_{\mathcal{F}}$. See detailed explanations in Section. 3.

(the structure of objective and constraints). "Diversity" implies that the generation method should be capable of catering to different problem formulations and encompassing extreme cases such as large dynamic ranges or degeneracy (Gamrath et al., 2020). Existing methods for MILP generation fall short of fulfilling the criteria above: Some are tailored to specific problems (e.g., knapsack (Hill et al., 2011) and quadratic assignment (Drugan, 2013)), requiring substantial expert effort for domain knowledge, hence struggling to generalize across different problems and failing in diversity; The others sample new instances in an embedding space by manipulating certain statistics (Smith-Miles & Bowly, 2015; Bowly et al., 2020; Bowly, 2019). The latter methods, which model MILPs' coefficients with simple distributions such as Gaussian distributions, generate instances with very limited structural characters, leading to not being representative enough.

With this in mind, we introduce DIG-MILP, a deep generative framework for MILP based on variational auto-encoder (VAE) (Kingma & Welling, 2013; Kipf & Welling, 2016). By employing deep neural networks (NNs) to extract the structural information, DIG-MILP enables the generation of "representative" data that resembles the original samples without expert knowledge. DIG-MILP leverages the MILP duality theories to ensure the feasibility and boundedness of each generated instance by controlling its primal format and the dual format of its linear relaxation having at least a feasible solution, which achieves the "authenticity" of the generated data. Moreover, any feasible-bounded MILP is inside the generation space of DIG-MILP, meeting the demand for "diversity". An illustration of DIG-MILP's generation strategy is shown in Figure. 1. Recognizing the limited original data along with the requirements on scalability and numerical precision in MILP generation, instead of generating from scratch, DIG-MILP iteratively modifies parts of existing MILPs, allowing control on the degree of structural similarity towards the original data.

We conduct two downstream tasks to validate the quality and novelty of DIG-MILP-generated instances, corresponding to the motivation of data generation in industry and in academia respectively. Specifically, the first task involves MILP problem sharing for solver hyper-parameter tuning without publishing original data. Across four distinct problems, the solution time of solver SCIP (Bestuzheva et al., 2021) exhibits a highly positive correlation between the DIG-MILP-generated instances and the original data w.r.t. different hyper-parameter sets. The other task is envisioned as data augmentation, where the generated instances assist in training NNs to predict the optimal objective values for MILP problems (Chen et al., 2023). Models trained on datasets augmented with DIG-MILP-generated instances demonstrate enhanced generalization capabilities.

## 2  RELATED WORK

In the following, we discuss works on MILP generation. In light of Hooker's proposals (Hooker, 1994; 1995), research on MILP generation diverges into two paths. The first focuses on leveraging expert domain knowledge to create generators for specific problems such as set covering (Balas & Ho, 1980), traveling sales person (Pilcher & Rardin, 1992; Vander Wiel & Sahinidis, 1995), graph colouring (Culberson, 2002), knapsack (Hill et al., 2011), and quadratic assignment (Drugan, 2013). This specificity causes poor generalization across different problems and thus fails diversity.

In contrast, the second path aims at generating general MILPs. Asahiro et al. (1996) propose to generate completely random instances, which is inadequate for producing instances with specific distributional features (Hill & Reilly, 2000). Bowly (2019); Bowly et al. (2020) attempt to sample feasible instances similar to target data by manually controlling distributions in an embedding space. The formulation used in (Bowly, 2019) to guarantee feasibility is similar to our method, however, its manual feature extraction and statistic control by simple distributions leads to instances with too limited structural characteristics to be representative enough. Inspired by Bowly (2019), DIG-MILP generates instances from the solution space and uses DNNs to dig out more details, aiming to delineate the structural attributes more precisely.

## 3    METHODOLOGY

We start by providing a preliminary background on MILP generation. Subsequently, we discuss the theoretical foundation based on which DIG-MILP's generation strategy ensures the feasibility and boundedness of its generated instances. Finally, we delve into the training and inference process of DIG-MILP along with its neural network architecture.

### 3.1    PRELIMINARIES

Given a triplet of coefficient matrix $\boldsymbol{A} \in \mathbb{R}^{m \times n}$, right-hand side constant $\boldsymbol{b} \in \mathbb{R}^m$, and objective coefficient $\boldsymbol{c} \in \mathbb{R}^n$, an MILP is defined as:

$$\textbf{MILP}(\boldsymbol{A}, \boldsymbol{b}, \boldsymbol{c}): \quad \max_{\boldsymbol{x}} \boldsymbol{c}^\top \boldsymbol{x}, \quad \text{s.t. } \boldsymbol{A}\,\boldsymbol{x} \le \boldsymbol{b},\ \boldsymbol{x} \in \mathbb{Z}_{\ge 0}^n. \tag{1}$$

To solve MILP is to identify a set of non-negative integer variables that maximize the objective function while satisfying a series of linear constraints. Merely finding a set of feasible solutions to such a problem could be NP-hard. Within the entire MILP space $\mathcal{H} = \{[\boldsymbol{A}, \boldsymbol{b}, \boldsymbol{c}] : \boldsymbol{A} \in \mathbb{R}^{m \times n}, \boldsymbol{b} \in \mathbb{R}^m, \boldsymbol{c} \in \mathbb{R}^n\}$, the majority of MILP problems are infeasible or unbounded. However, *In real-world business scenarios, MILPs derived from practical issues are often expected to be feasible, bounded, and yield an optimal solution*[1], otherwise the modeling for the practical problem would be meaningless. Therefore, we are particularly interested in MILPs from the following space that corresponds to feasible-bounded instances only: [2]

$$\mathcal{H}' := \{[\boldsymbol{A}, \boldsymbol{b}, \boldsymbol{c}] : \boldsymbol{A} \in \mathbb{Q}^{m \times n}, \boldsymbol{b} \in \mathbb{Q}^m, \boldsymbol{c} \in \mathbb{Q}^n \text{ and } \text{MILP}(\boldsymbol{A}, \boldsymbol{b}, \boldsymbol{c}) \text{ is feasible and bounded.}\}.$$

Suppose a target MILP dataset $D$ that models a particular business scenario is sampled from a distribution $\mathcal{D}_{\mathcal{H}'}(\boldsymbol{A}, \boldsymbol{b}, \boldsymbol{c})$ defined on $\mathcal{H}'$, the task of MILP instance generation is to approximate the distribution $\mathcal{D}_{\mathcal{H}'}$ and sample novel MILP instances from it.

### 3.2    DIG-MILP WITH FEASIBILITY GUARANTEE

An intuitive idea for MILP generation is to directly sample $[\boldsymbol{A}, \boldsymbol{b}, \boldsymbol{c}]$ from $\mathcal{D}_{\mathcal{H}'}$, which is practically hard to implement as it's hard to guarantee the generated instance to be feasible-bounded.

According to MILP duality theories, we observe that as long as DIG-MILP could ensure that a generated instance's primal format MILP$(\boldsymbol{A}, \boldsymbol{b}, \boldsymbol{c})$ and the dual format of its linear relaxation DualLP$(\boldsymbol{A}, \boldsymbol{b}, \boldsymbol{c})$ (as defined in Equation. 2) both have at least one set of feasible solutions, then the newly generated instance will be guaranteed to be feasible-bounded (as proved in Theorem. 1).

$$\textbf{DualLP}(\boldsymbol{A}, \boldsymbol{b}, \boldsymbol{c}): \quad \max_{\boldsymbol{y}} \boldsymbol{b}^\top \boldsymbol{y}, \quad \text{s.t. } \boldsymbol{A}^\top \boldsymbol{y} \ge \boldsymbol{c},\ \boldsymbol{y} \ge 0, \tag{2}$$

To guarantee the existence of feasible solutions to both problems, inspired by (Bowly, 2019), we propose to sample the instances from another space $\mathcal{F}$, where

$$\mathcal{F} := \{[\boldsymbol{A}, \boldsymbol{x}, \boldsymbol{y}, \boldsymbol{s}, \boldsymbol{r}] : \boldsymbol{A} \in \mathbb{Q}^{m \times n}, \boldsymbol{x} \in \mathbb{Z}_{\ge 0}^n, \boldsymbol{y} \in \mathbb{Q}_{\ge 0}^m, \boldsymbol{s} \in \mathbb{Q}_{\ge 0}^n, \boldsymbol{r} \in \mathbb{Q}_{\ge 0}^m\}. \tag{3}$$

---

[1]Definitions of boundedness, feasibility, and optimal solution of MILP in Definition. 1 2 3 in the appendix.

[2]Narrowing from the real domain to the rational domain is common in MILP studies to avoid cases where an MILP is feasible and bounded but lacks an optimal solution Schrijver (1998). For example, $\min \sqrt{3}x_1 - x_2$, s.t. $\sqrt{3}x_1 - x_2 \ge 0, x_1 \ge 1, \boldsymbol{x} \in \mathbb{Z}_{\ge 0}^2$. No feasible solution has objective equal to zero, but there are feasible solutions with objective arbitrarily close to zero.

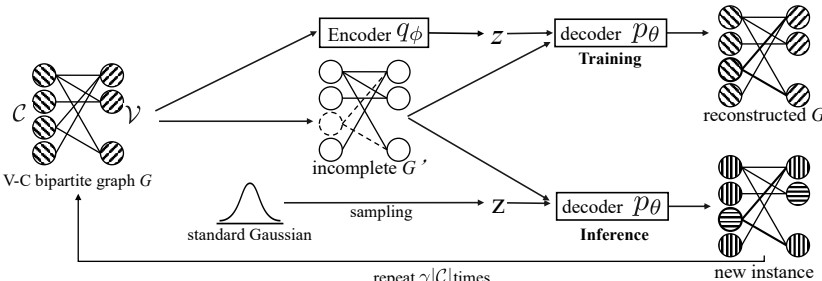

Figure 2: The training and inference pipeline of DIG-MILP. In each training step, DIG-MILP removes a random constraint node, its connected edges, along with the solution and slack features on all the nodes, resulting in an incomplete graph $G'$. The training objective of DIG-MILP is to reconstruct $G$ from $G'$ and $z$ sampled by the encoder $q_\phi$. As to inference, DIG-MILP employs an auto-regressive approach, generating new instances by iteratively modifying the existing MILPs.

$\mathcal{F}$ defines an alternative space to represent feasible-bounded MILPs, with each element $[A, x, y, s, r]$ consisting of the coefficient matrix $A$ along with a set of feasible solutions $x, y$ to MILP$(A, b, c)$ and DualLP$(A, b, c)$, respectively, where $b, c$ are determined by the corresponding slacks $s, r$ via the equalities defined in Equation. 4. By leveraging this idea, DIG-MILP aims to learn a distribution $\mathcal{G}_\mathcal{F}$ over the space of $\mathcal{F}$ to sample $[A, x, y, s, r]$, which can be further transformed into $[A, b, c]$ that defines an MILP problem based on Equation. 4.

$$\textbf{Slack Variables:} \quad Ax + r = b, A^\top y - s = c, \quad \text{where } r \in \mathbb{Q}_{\geq 0}^m, s \in \mathbb{Q}_{\geq 0}^n \tag{4}$$

Such a generation strategy offers theoretical guarantees on the boundedness and feasibility of the generated instances, ensuring the "authenticity" of the produced data. Moreover, all the feasible and bounded MILPs in $\mathcal{H}'$ correspond to at least a tuple $[A, x, y, s, r]$. Therefore, this procedure also offers theoretical assurances for the capability to produce "diverse" instances. These points are formally stated in Theorem. 1. See detailed proof in A.1 in the appendix.

**Theorem 1** (**Boundedness and Feasibility Guarantee of DIG-MILP**). *DIG-MILP guarantees to produce feasible-bounded MILP instances only, and any feasible-bounded MILP could be generated by DIG-MILP. In other words, it holds that $\mathcal{H}' = \Big\{ [A, b, c] : b = Ax + r, c = A^\top y - s, [A, x, y, s, r] \in \mathcal{F} \Big\}$.*

### 3.3 GENERATION PROCESS AND ARCHITECTURE

Having shown the equivalence between sampling from space $\mathcal{F}$ and $\mathcal{H}'$, we then present how DIG-MILP learns a distribution $\mathcal{G}_\mathcal{F}$ to sample $[A, x, y, x, r]$ from. We encode each $[A, x, y, s, r]$ as a variable-constraint (VC) bipartite graph $G(\mathcal{V}, \mathcal{C}, \mathcal{E})$: On side $\mathcal{V}$, each node in $\{v_1, ..., v_m\}$ corresponds to a variable, while on $\mathcal{C}$ side, each node in $\{c_1, ..., c_m\}$ represents a constraint. Edges in $\mathcal{E}$ connect constraints to variables according to the non-zero entries in the coefficient matrix $A$, implying that $A$ serves as the adjacency matrix of graph $G$. The input features of nodes and edges are detailed in Table. 1. With this graph representation, we transform the MILP generation challenge into a graph generation task. DIG-MILP iteratively modifies part of the original graph to produce new graphs.

**Generation pipeline** We display the training and inference pipeline in Figure. 2. As illustrated in Algorithm. 1, on each training step of DIG-MILP, we randomly select and remove a constraint node $c_i$ (corresponding to the $i$-th constraint) from the bipartite graph, along with all its connected edges $\mathcal{E}_G(c_i)$. Concurrently, we erase the features of the solution space $x, y, s, r$ on all the nodes, resulting in an incomplete graph $G'(\mathcal{C} \backslash c_{i-y,s}; \mathcal{V}_{-x,r}; \mathcal{E} \backslash \mathcal{E}_G(c_i))$. The training objective is to learn DIG-MILP to reconstruct $G$ from the given $G'$ by maximizing the log likelihood:

Table 1: The input encoding into $G$ from MILP.

| object | feature |
|---|---|
| constraint-nodes: | all 0's |
| | $y = [y_1, ..., y_m]^\top$ |
| $\mathcal{C} = \{c_1...c_m\}$ | $r = [r_1, ..., r_m]^\top$ |
| variable-nodes: | all 1's |
| | $x = [x_1, ..., x_n]^\top$ |
| $\mathcal{V} = \{v_1...v_n\}$ | $s = [s_1, ..., s_n]^\top$ |
| edge $\mathcal{E}$ | non-zero weights in $A$ |

$$\arg\max_{\theta,\phi} \mathbb{E}_{G\sim D}\mathbb{E}_{G'\sim p(G'|G)} \log \mathbb{P}(G|G';\theta,\phi), \tag{5}$$

where $p(G'|G)$ refers to randomly removing structures along with features to produce the incomplete graph, $\theta$ and $\phi$ denote the NN parameters. To address the dependency issues and foster diversity into generation, we adhere to the standard procedure in VAEs (Kingma & Welling, 2013; Kipf & Welling, 2016) by introducing a latent variable $z = [z_1, ..., z_{m+n}]$ with the assumption that $z$ is independent with $G'$. Utilizing the principles of the variational evidence lower bound (ELBO), we endeavor to maximize the training objective through the optimization of the ensuing loss function:

$$\min_{\theta,\phi} \mathcal{L}_{\theta,\phi} = \mathbb{E}_{G\sim D}\mathbb{E}_{G'\sim p(G'|G)} \left[ \alpha\mathbb{E}_{z\sim q_\phi(z|G)}[-\log p_\theta(G|G',z)] + \mathcal{D}_{KL}[q_\phi(z|G)\|\mathcal{N}(0,I)] \right], \tag{6}$$

where the decoder parameterized by $\theta$ is to adeptly reconstruct graph $G$ based on the latent variables $z$ and the incomplete graph $G'$; the encoder parameterized by $\phi$ is to depict the posterior distribution of $z$ which is required to align with the prior standard Gaussian. The hyper-parameter $\alpha$ functions as a balancing factor between the two parts of the loss. See detailed derivation of the loss in A.2 in the appendix. During training, DIG-MILP modifies only one constraint of the data at a time. In the inference phase, the graph rebuilt after removing a constraint can be fed back as an input, allowing iterative modifications to the original data. The number of iterations controls the degree of structural similarity to the original problem. The inference procedure is shown in Algorithm. 2, where $\gamma|\mathcal{C}$ denotes the number of iterations to remove a constraint.

---

**Algorithm 1** DIG-MILP Training

**Require:** : dataset $D$, epoch $N$, batch size $B$
1: Solve MILPs for $\{[x, y, s, r]\}$ over $D$
2: Encode MILPs into graphs $\{G(\mathcal{V}, \mathcal{C}, \mathcal{E})\}$
3: **for** epoch=1,...,N **do**
4:      Allocate empty batch $\mathcal{B} \leftarrow \emptyset$
5:      **for** idx=1,...,$B$ **do**
6:          $G \sim D$; $G' \sim p(G'|G)$
7:          $\mathcal{B} \leftarrow \mathcal{B} \cup \{(G, G')\}$
8:          Encode $z \sim q_\phi(z|G)$
9:          Decode $G \sim p_\theta(G|G', z)$
10:         Calculate $\mathcal{L}_{\theta,\phi}(G, G')$
11:      **end for**
12:      $\mathcal{L}_{\theta,\phi} \leftarrow \frac{1}{B}\sum_{(G,G')\in\mathcal{B}} \mathcal{L}_{\theta,\phi}(G, G')$
13:      Update $\phi, \theta$ by minimizing $\mathcal{L}_{\theta,\phi}$
14: **end for**
15: **return** $\theta, \phi$

**Algorithm 2** DIG-MILP Inference

**Require:** : dataset $D$, batch size $B$, constraint replace rate $\gamma$
1: Solve MILPs for $\{[x, y, s, r]\}$ over $D$
2: Encode MILPs into graphs $\{G(\mathcal{V}, \mathcal{C}, \mathcal{E})\}$
3: Allocate empty batch $\mathcal{B} \leftarrow \emptyset$
4: **for** id=1,...,$B$ **do**
5:      $G \sim D$
6:      **for** t=1,...,$\gamma|\mathcal{C}|$ **do**
7:          $G' \sim p(G'|G)$
8:          $z \sim \mathcal{N}(0, I)$
9:          Decode $\tilde{G} \sim p_\theta(\tilde{G}|G', z)$
10:         $G \leftarrow \tilde{G}$
11:      **end for**
12:      $\mathcal{B} \leftarrow \mathcal{B} \cup G$
13: **end for**
14: **return** new instance batch $\mathcal{B}$

---

**Neural Network Architecture** For both the encoder and decoder, we employ the same bipartite graph neural network (GNN) as delineated in (Gasse et al., 2019) as the backbone. The encoder encodes the graph into the distribution of the latent variable $z$, as depicted in the following equation:

$$q_\phi(z|G) = \prod_{u\in\mathcal{C}\cup\mathcal{V}} q_\phi(z_u|G), \qquad q_\phi(z_u|G) = \mathcal{N}(\mu_\phi(h_u^G), \Sigma_\phi(h_u^G)), \tag{7}$$

where $z_u$ is conditionally independent with each other on $G$, $h^G = \text{GNN}_\phi(G)$ denotes the node embeddings of $G$ outputted by the encoder backbone, $\mu_\phi$ and $\Sigma_\phi$ are two MLP layers that produce the mean and variance for the distribution of $z$. The decoder connects seven parts conditionally independent on the latent variable and node representations, with detailed structure as follows:

$$p_\theta(G|G', z) = p_\theta(d_{c_i}|h_{c_i}^{G'}, z_{c_i}) \cdot \prod_{u\in\mathcal{V}} p_\theta(e(c_i, u)|h_\mathcal{V}^{G'}, z_\mathcal{V}) \cdot \prod_{u\in\mathcal{V}:e(c_i,u)=1} p_\theta(w_{c_i}|h_\mathcal{V}^{G'}, z_\mathcal{V})$$
$$\cdot \prod_{u\in\mathcal{C}} p_\theta(y_u|h_\mathcal{C}^{G'}, z_\mathcal{C})p_\theta(r_u|h_\mathcal{C}^{G'}, z_\mathcal{C}) \cdot \prod_{u\in\mathcal{V}} p_\theta(x_u|h_\mathcal{V}^{G'}, z_\mathcal{V})p_\theta(s_u|h_\mathcal{V}^{G'}, z_\mathcal{V}), \tag{8}$$

where $z_\mathcal{C}, h_\mathcal{C}^{G'}$ denotes the latent variable and node representations on side $\mathcal{C}$ outputted by the decoder backbone, while $z_\mathcal{V}, h_\mathcal{V}^{G'}$ signifies those on side $\mathcal{V}$; $d_{c_i}$ predicts the degree of the deleted node

Table 2: Datasets Meta-data . For CVS and IIS, 'training' (non-bold) instances are for DIG-MILP or downstream model training, 'testing' (bold) instances are used in downstream testing only.

| | SC | CA | CVS | | | | | IIS | |
| --- | --- | --- | --- | --- | --- | --- | --- | --- | --- |
| # data | 1000 | 1000 | training | | | testing | | training | testing |
| | | | cvs08r139-94 | cvs16r70-62 | cvs16r89-60 | **cvs16r106-72** | **cvs16r128-89** | iis-glass-cov | **iis-hc-cov** |
| # variable | 400 | 300 | 1864 | 2112 | 2384 | 2848 | 3472 | 214 | 297 |
| # constraint | 200 | ∼10^2 | 2398 | 3278 | 3068 | 3608 | 4633 | 5375 | 9727 |
| difficulty | easy | easy | hard | | | | | hard | |

$c_i$; $e(c_i, \cdot)$ denotes the probability of an edge between $c_i$ and a node on side $\mathcal{V}$; $w_{c_i}$ is the edge weights connected with $c_i$; $\boldsymbol{x}, \boldsymbol{y}, \boldsymbol{s}, \boldsymbol{r}$ are value of the solution and slacks. We use separate layers of MLP to model each part's prediction as a regression task. We optimize each part of the decoder with the Huber Loss (Huber, 1992). See Section. B.1 in the appendix for more details.

# 4 NUMERICAL EVALUATIONS

In this section, we first delineate the experimental setup. Then we calculate the structural statistical similarity between generated and original instances. Subsequently, we evaluate DIG-MILP with two downstream tasks: *(i)* MILP data sharing for solver tuning and *(ii)* MILP data augmentation for ML model training.

## 4.1 SETTINGS

**Datasets:** We perform DIG-MILP on four MILP datasets, encompassing scenarios involving simple and complex instances, a mix of small and large problem scale, varying instance quantities, and generation/collection from both synthetic and real-world sources. Specifically, we include two manually generated datasets, namely the set covering (SC) and the combinatorial auctions (CA), following the generation methodologies outlined in (Gasse et al., 2019). The remaining two datasets, namely CVS and IIS, are from the MIPLIB2017 benchmark (Gleixner et al., 2021)[3], which comprises challenging instances from a large pool of problem-solving contexts. CVS pertains to the capacitated vertex separator problem on hypergraphs, while IIS mirrors real-world scenarios and resembles the set covering problems. Details are elaborated in Table. 2. It's worth emphasizing that for CVS and IIS, we exclusively employ the 'training' data during the training of DIG-MILP and all downstream models. The 'testing' data is used only for downstream task evaluation.

**Downstream Tasks:** We devise two downstream applications, tailored to address distinct motivations. One motivation pertains to generating and sharing data that can substitute target instances. The other motivation involves data augmentation for better training ML models.

*(S1): Data Sharing for Solver Configuration Tuning* We simulate the process where clients utilize DIG-MILP to generate new instances and hand over to companies specializing in MILP solver tuning. In particular, we calculate the Pearson positive correlation of the solution times required by the SCIP (Bestuzheva et al., 2021) solver between the generated examples and the original testing data across various hyper-parameter configurations. Should the solution time consistently demonstrate a positive correlation between the original and generated problems across varied parameter settings, it implies a consistent level of the effectiveness on the original and new instances under the same parameter configuration, which facilitates sharing data for parameter tuning.

*(S2): Optimal Value Prediction via ML* Following the settings presented in (Chen et al., 2023), this supervised regression task employs GNNs to express the optimal value of the objective function in an MILP. We utilize newly generated instances as a means of augmentation to formulate training datasets for ML models. For more detailed implementation, see B.5 in the appendix.

**Solvers and Baselines:** We use the open source solver SCIP (Bestuzheva et al., 2021) with its Python interface, namely PySCIPOpt (Maher et al., 2016b) for all the experiments. We consider two approaches as our baselines. The first, named 'Bowly', aligns with Bowly (2019) that generates MILP instances from scratch by sampling in an embedding space based on manually designed distributions. The second baseline 'random' employs identical NN architectures to DIG-MILP but randomizes the network's outputs, further validating the importance and efficacy of model training. For more implementation details of the baselines, please refer to B.2 in the appendix.

---

[3]`https://miplib.zib.de/tag_benchmark.html`

Table 3: The similarity score ↑ between the original and generated data .

| constraint replace rates $\gamma$ | | - | 0.01 | 0.05 | 0.10 | 0.20 | 0.50 |
|---|---|---|---|---|---|---|---|
| | Bowly | 0.337 | - | - | - | - | - |
| SC | random | - | 0.701 | 0.604 | 0.498 | 0.380 | 0.337 |
| | ours | - | **0.856** | **0.839** | **0.773** | **0.652** | **0.570** |
| | Bowly | 0.386 | - | - | - | - | - |
| CA | random | - | 0.630 | 0.566 | 0.508 | 0.432 | 0.306 |
| | ours | - | **0.775** | **0.775** | **0.768** | **0.733** | **0.630** |

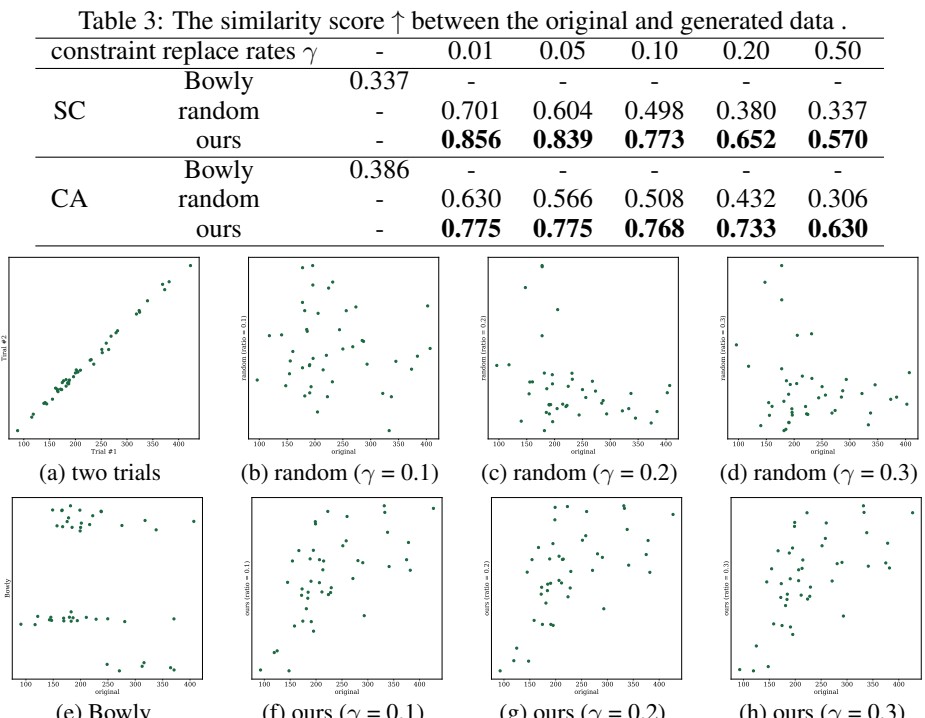

| (a) two trials | (b) random ($\gamma = 0.1$) | (c) random ($\gamma = 0.2$) | (d) random ($\gamma = 0.3$) |
|---|---|---|---|
| (e) Bowly | (f) ours ($\gamma = 0.1$) | (g) ours ($\gamma = 0.2$) | (h) ours ($\gamma = 0.3$) |

Figure 3: The solution time (second) of SCIP on CVS with 45 different hyper-parameter sets.

## 4.2 RESULTS AND ANALYSIS

### 4.2.1 STATISTICAL CHARACTERISTICS OF THE GENERATED INSTANCES

We compare the statistical metrics between the generated instances and the original instances on the SC and CA datasets. We do not calculate the statistics on the CVS and IIS due to their limited size that prevents meaningful statistical comparisons. We count nine statistic metrics in total, see Table. B.3 in the appendix for details. The similarity score is derived from the Jensen-Shannon (JS) divergence (the lower the better) between each metric of the generated and original data, as shown in Table. 3. 'Bowly' shows the least similarity. As the the constraint replacement ratio $\gamma$ increases from 0.01 to 0.50, the table shows a decreasing similarity between new and original instances for both DIG-MILP and 'random', aligning with our expectation of controlling structural similarity by adjusting the number of constraint nodes to replace. Instances generated by DIG-MILP more closely mirror the target data in structural statistical metrics across all $\gamma$. For detailed calculations of the similarity score and the specific values of each statistic metric, see B.3 and C.1 in the appendix.

### 4.2.2 DOWNSTREAM TASK #1: DATA SHARING FOR SOLVER CONFIGURATION TUNING

We conduct experiments on all the four datasets. SCIP boasts an extensive array of parameters, rendering a tuning across the entire range impractical. Therefore, we adopt the reduced parameter space consistent with mainstream research on SCIP solver tuning (Hutter et al., 2011; Lindauer & Hutter, 2018; Lindauer et al., 2022). See Table. 12 in the appendix for detailed parameter space selection. We employ random seed $0 - 44$ to generate 45 distinct parameter configurations. To validate the impact of randomness on SCIP, we initiate two independent trials on the same original testing data and compare the Pearson score of solution time. As illustrated in the diagonal of Table. 4, it clearly demonstrates a very high positive correlation for two independent trials on the same data. For subsequent experiments, each is run three times independently, with results averaged to mitigate randomness effects. We then compare the correlation of solution time on the original data across different datasets, as presented in the upper triangle of Table. 4. We observe a certain degree of positive correlation between synthetic datasets SC and CA, as well as between MIPLIB datasets CVS and IIS, which reveals that the effectiveness of parameters may naturally exhibit some degree of generalization across similar problems. However, the correlation between synthetic and MIPLIB datasets tends to be much lower, underscoring the necessity of generating new instances for solver

Table 4: The Pearson correlation coefficient ('r') and the significance value ('p') of the SCIP solution time under 45 different hyper-parameters on dataset-pairs.

|  |  | SC | CA | CVS | IIS |
|---|---|---|---|---|---|
| SC | r | 0.732 | 0.599 | 0.115 | 0.088 |
|  | p | 1.058e-8 | 1.351e-5 | 0.449 | 0.561 |
| CA | r | - | 0.952 | 0.021 | 0.092 |
|  | p | - | 0.762e-24 | 0.890 | 0.545 |
| CVS | r | - | - | 0.997 | 0.550 |
|  | p | - | - | 4.723e-53 | 9.033e-5 |
| IIS | r | - | - | - | 0.988 |
|  | p | - | - | - | 1.563e-36 |

Table 5: The Pearson correlation coefficient ('r') and the significance value ('p') of the SCIP solution time between generated data and the original testing data under 45 different hyper-parameters on the SC, CA, CVS, and IIS problem.

|  |  | CA | | | SC | | | CVS | | | IIS | | |
|---|---|---|---|---|---|---|---|---|---|---|---|---|---|
| Bowly | r | -0.048 | | | 0.683 | | | -0.158 | | | 0.292 | | |
|  | p | 0.751 | | | 2.295e-7 | | | 0.298 | | | 0.051 | | |
| ratio | | 0.10 | 0.20 | 0.30 | 0.10 | 0.20 | 0.30 | 0.10 | 0.20 | 0.30 | 0.10 | 0.20 | 0.30 |
| random | r | 0.723 | 0.563 | 0.515 | 0.542 | 0.568 | 0.609 | -0.085 | -0.337 | -0.201 | 0.114 | 0.182 | 0.149 |
|  | p | 1.971e-8 | 5.522e-5 | 2.942e-4 | 1.174e-4 | 4.535e-5 | 9.028e-6 | 0.578 | 0.023 | 0.184 | 0.452 | 0.228 | 0.327 |
| ours | r | **0.728** | **0.771** | **0.780** | **0.747** | **0.717** | **0.665** | **0.609** | **0.590** | **0.607** | **0.542** | **0.300** | **0.551** |
|  | p | **1.446e-8** | **5.371e-10** | **2.544e-10** | **3.646e-9** | **2.908e-8** | **6.353e-7** | **8.834e-6** | **1.986e-5** | **9.581e-6** | **1.187e-4** | **0.044** | **8.497e-5** |

tuning on specific problems. Finally, we compare the positive correlation of solution time between the generated instances and the original testing instances of the same datasets, as shown in Table. 5. Across all four datasets, the DIG-MILP-generated instances, exhibit the highest correlation with the testing data compared to the baselines, with the lowest p-value of significance. On the MIPLIB test set, DIG-MILP-generated instances exhibits a slightly lower correlation, primarily due to the very few samples in these datasets. We visualize the correlation of solution time between the original testing data and the generated data on the CVS in Figure. 3. More detailed implementation and the visualization of the other datasets can be found in B.4 and Figure. 4-7 in the appendix.

### 4.2.3 DOWNSTREAM TASK #2: OPTIMAL VALUE PREDICTION VIA MACHINE LEARNING

We conduct experiments for the second downstream task on all four datasets.

**Set Covering (SC)** One of the hyper-parameters of the SC instances is 'density', representing the number of sets to be covered within a constraint. The training set (for both DIG-MILP and the downstream predictor) comprises data with densities ranging from 0.15 to 0.35 only. We not only present test sets for each in-distribution density (0.15 to 0.35) but also design the test sets with densities falling within the unexplored range of 0.03 to 0.10, to reflect the predictor's ability to generalize across distribution shift. The relative mean squared error (MSE) values of the models' predictions are presented in Table. 6. In the first part (Datasets #1-#8), we curate datasets with a fixed training set size of 1000. Dataset #1 consist of 1000 original data, dataset #2 generates instance via the 'Bowly', #3 uses the 'random' baseline. Datasets #4-#8 comprise a combination of 500 original instances and 500 DIG-MILP-generated instances, with varying constraint node replacement ratios $\gamma$ ranging from 0.01 to 0.50. Models trained exclusively on in-distribution data exhibit superior fitting and predictive accuracy within the in-distribution test sets. However, models trained on a combination of original and DIG-MILP-generated instances display significantly enhanced prediction accuracy on out-of-distribution testing data. We attribute this phenomenon to the increased structural and label diversity in the newly generated instances, mitigating over-fitting on in-distribution data and consequently bolstering the model's cross-distribution capabilities. It's worth noting that 'Bowly' or 'random' neither enhances the model's in-distribution nor out-of-distribution performance. We believe this is due to the less precise representation of the target distribution by the manually-designed 'Bowly' baseline and the excessively high randomness in 'random', causing the generated instances to deviate substantially from the original problems in both solution space and structure. In the second part (Datasets #9-#14), we investigate the impact of progressively incorporating DIG-MILP-generated instances into the dataset, initially starting with 500 original instances. We observe a consistent improvement in model performance with the gradual inclusion of additional newly generated instances, with peak performance achieved when augmenting the dataset with 500 newly generated instances.

Table 6: The relative mean square error (MSE) of the optimal objective value task on the set covering (SC) problem. The $500$ original instances in training dataset $\#2 - \#14$ are identical.

| dataset | #original | #generated | replace ratio | out-of-distribution | | | | in-distribution | | | | |
|---|---|---|---|---|---|---|---|---|---|---|---|---|
| | | | | **0.03** | **0.04** | **0.05** | **0.10** | 0.15 | 0.20 | 0.25 | 0.30 | 0.35 |
| 1 | 1000 | 0 | - | 0.792 | 0.640 | 0.488 | **0.022** | **0.009** | **0.009** | **0.010** | **0.011** | **0.015** |
| 2 | 500 | 500 (Bowly) | - | 3.498 | 17.671 | 43.795 | 81.408 | 0.037 | 0.052 | 0.052 | 0.065 | 0.045 |
| 3 | 500 | 500 (random) | 0.10 | 0.449 | 4.176 | 12.624 | 86.592 | 0.048 | 0.064 | 0.053 | 0.069 | 0.045 |
| 4 | 500 | 500 (DIG-MILP) | 0.01 | 0.505 | 0.280 | 0.142 | 0.032 | 0.032 | 0.040 | 0.044 | 0.044 | 0.040 |
| 5 | 500 | 500 (DIG-MILP) | 0.05 | 0.575 | 0.329 | 0.155 | 0.080 | 0.036 | 0.044 | 0.046 | 0.056 | 0.056 |
| 6 | 500 | 500 (DIG-MILP) | 0.10 | **0.362** | **0.141** | **0.045** | 0.065 | 0.017 | 0.012 | 0.012 | 0.010 | 0.015 |
| 7 | 500 | 500 (DIG-MILP) | 0.20 | 0.625 | 0.418 | 0.265 | 0.034 | 0.059 | 0.083 | 0.077 | 0.099 | 0.069 |
| 8 | 500 | 500 (DIG-MILP) | 0.50 | 0.884 | 0.822 | 0.769 | 0.285 | 0.017 | 0.025 | 0.033 | 0.047 | 0.032 |
| 9 | 500 | 0 | - | 0.868 | 0.758 | 0.637 | 0.072 | **0.016** | 0.014 | 0.014 | 0.017 | 0.027 |
| 10 | 500 | 50 (DIG-MILP) | 0.10 | 0.693 | 0.497 | 0.327 | 0.031 | 0.035 | 0.039 | 0.046 | 0.039 | 0.052 |
| 11 | 500 | 100 (DIG-MILP) | 0.10 | 0.603 | 0.361 | 0.179 | 0.096 | 0.031 | 0.033 | 0.038 | 0.042 | 0.038 |
| 12 | 500 | 200 (DIG-MILP) | 0.10 | 0.628 | 0.396 | 0.215 | 0.086 | 0.038 | 0.035 | 0.039 | 0.043 | 0.039 |
| 13 | 500 | 500 (DIG-MILP) | 0.10 | **0.362** | **0.141** | **0.045** | **0.065** | 0.017 | **0.012** | **0.012** | **0.010** | **0.015** |
| 14 | 500 | 1000 (DIG-MILP) | 0.10 | 0.473 | 0.211 | 0.063 | 0.339 | 0.013 | 0.014 | 0.014 | 0.014 | 0.024 |

Table 7: The relative mean square error (MSE) of the optimal objective value task on the combinatorial auction (CA) problem. The $500$ original instances in training dataset $\#2 - \#14$ are identical.

| dataset | #original | #generated | replace ratio | in-distribution | | | out-of-distribution | | | |
|---|---|---|---|---|---|---|---|---|---|---|
| | | | | 40/200 | 60/300 | 80/400 | **100/500** | **120/600** | **140/700** | **160/800** |
| 1 | 1000 | 0 | - | 0.246 | **0.003** | 0.060 | 0.155 | 0.239 | 0.312 | 0.379 |
| 2 | 500 | 500 (Bowly) | - | **0.202** | 0.004 | 0.080 | 0.183 | 0.272 | 0.346 | 0.410 |
| 3 | 500 | 500 (random) | 0.10 | 0.242 | 0.006 | 0.077 | 0.179 | 0.269 | 0.347 | 0.409 |
| 4 | 500 | 500 (DIG-MILP) | 0.01 | 0.346 | 0.008 | 0.043 | 0.131 | 0.219 | 0.292 | 0.359 |
| 5 | 500 | 500 (DIG-MILP) | 0.05 | 0.345 | 0.009 | 0.041 | 0.125 | 0.211 | 0.284 | 0.352 |
| 6 | 500 | 500 (DIG-MILP) | 0.10 | 0.385 | 0.015 | 0.036 | **0.118** | **0.201** | **0.276** | **0.340** |
| 7 | 500 | 500 (DIG-MILP) | 0.20 | 0.428 | 0.019 | **0.035** | 0.116 | 0.203 | 0.275 | 0.344 |
| 8 | 500 | 500 (DIG-MILP) | 0.30 | 0.381 | 0.012 | 0.040 | 0.126 | 0.215 | 0.289 | 0.356 |
| 9 | 500 | 500 (DIG-MILP) | 0.50 | 0.398 | 0.014 | 0.035 | 0.117 | 0.203 | 0.276 | 0.344 |
| 10 | 500 | 0 | - | **0.216** | **0.004** | 0.068 | 0.165 | 0.249 | 0.324 | 0.388 |
| 11 | 500 | 50 (DIG-MILP) | 0.10 | 0.382 | 0.006 | 0.040 | 0.130 | 0.218 | 0.293 | 0.361 |
| 12 | 500 | 100 (DIG-MILP) | 0.10 | 0.446 | 0.014 | **0.031** | 0.116 | 0.201 | 0.275 | 0.344 |
| 13 | 500 | 500 (DIG-MILP) | 0.10 | 0.385 | 0.015 | 0.036 | **0.118** | **0.201** | **0.276** | **0.340** |
| 14 | 500 | 1000 (DIG-MILP) | 0.10 | 0.359 | 0.009 | 0.039 | 0.126 | 0.212 | 0.285 | 0.351 |

**Combinatorial Auctions (CA)** One of the hyper-parameters for the CA is the number of bid/item pair, which determines the quantity of variables and constraints. Our training set exclusively comprises examples with bid/item values ranging from $40/200$ to $80/400$. With the setting similar to the SC, our testing set not only has in-distribution bid/item value pairs, but also introduces instances with bid/item values ranging from $40/200$ to $160/800$, allowing us to assess the model's ability of cross-scale generalization. The relative mean squared error (MSE) of the model's predictions is provided in Table. 7. The experiments are also divided into two parts. The first part (Datasets #1-#8) yields similar conclusions, where models trained solely on original data excel in fitting within in-distribution test sets, models trained on a mixture of half original and half DIG-MILP-generated instances perform better on test sets at scales never encountered during training (bid/item ranging from $100/500$ to $160/800$). This observation is attributed to the diversity introduced by the generated instances, in terms of both the problem structure and optimal objective labels, that prevents the models from over-fitting and thereby enhance their generalization across scales. Consistent with the SC, the second part demonstrates the impact of gradually increasing the new instances as training data and also achieves the peak performance with $500$ newly generated instances.

**CVS and IIS** Experiments on CVS and IIS show similar insights, see Appendix. C.2 for details.

## 5 CONCLUSION

This paper introduces DIG-MILP, a deep generative framework for MILP. Contrasting with conventional MILP generation techniques, DIG-MILP does not rely on domain-specific expertise. Instead, it employs DNNs to extract profound structural information from limited MILP data, generating "representative" instances. Notably, DIG-MILP guarantees the feasibility and boundedness of generated data, ensuring the data's "autheticity". The generation space of DIG-MILP encompasses any feasible-bounded MILP, providing it with the capability of generating "diverse" instances. Experiment evaluations highlights DIG-MILP's potential in (S1) MILP data sharing for solver hyperparameter tuning without publishing the original data and (S2) data augmentation to enhance the generalization capacity of ML models tasked with solving MILPs.

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

# A SUPPLEMENTARY THEORETICAL RESULTS

## A.1 PROOF OF THEOREM 1

Before the proof of Theorem 1, we first give the definition of boundedness, feasibility and optimal solutions of MILP, then we discuss the existence of optimal solutions of LP and MILP.

**Definition 1** (Feasibility of MILP). *An MILP$(\boldsymbol{A}, \boldsymbol{b}, \boldsymbol{c})$ is feasible if there exists an $\boldsymbol{x}$ such that all the constraints are satisfied: $\boldsymbol{x} \in \mathbb{Z}_{\geq 0}^n$, $\boldsymbol{A}\boldsymbol{x} \leq \boldsymbol{b}$. Such an $\boldsymbol{x}$ is named a feasible solution.*

**Definition 2** (Boundedness of MILP). *An MILP$(\boldsymbol{A}, \boldsymbol{b}, \boldsymbol{c})$ is bounded if there's an upper bound on $\boldsymbol{c}^\top \boldsymbol{x}$ across all feasible solutions.*

**Definition 3** (Optimal Solution for MILP). *A vector $\boldsymbol{x}^\star$ is recognized as an optimal solution if it's a feasible solution and it is no worse than all other feasible solutions: $\boldsymbol{c}^\top \boldsymbol{x}^\star \geq \boldsymbol{c}^\top \boldsymbol{x}$, given $\boldsymbol{x}$ is feasible.*

All LPs must fall into one of the following cases Bertsimas & Tsitsiklis (1997):

- Infeasible.

- Feasible but unbounded.

- Feasible and bounded. Only in this case, the LP yields an optimal solution.

However, general MILP will be much more complicated. Consider a simple example: $\min \sqrt{3}x_1 - x_2$, s.t. $\sqrt{3}x_1 - x_2 \geq 0, x_1 \geq 1, \boldsymbol{x} \in \mathbb{Z}_{\geq 0}^2$. No feasible solution has objective equal to zero, but there are feasible solutions with objective arbitrarily close to zero. In other words, *an MILP might be bounded but with no optimal solutions.* Such a pathological phenomenon is caused by the irrational number $\sqrt{3}$ in the coefficient. Therefore, we only consider MILP with rational data:

$$\boldsymbol{A} \in \mathbb{Q}^{m \times n}, \boldsymbol{b} \in \mathbb{Q}^m, \boldsymbol{c} \in \mathbb{Q}^m.$$

Such an assumption is regularly adopted in the research of MILP.

Without requiring $\boldsymbol{x}$ to be integral, equation 1 will be relaxed to an LP, named its *LP relaxation*:

$$\textbf{LP}(\boldsymbol{A}, \boldsymbol{b}, \boldsymbol{c}): \quad \max_{\boldsymbol{x}} \boldsymbol{c}^\top \boldsymbol{x}, \quad \text{s.t. } \boldsymbol{A}\,\boldsymbol{x} \leq \boldsymbol{b},\ \boldsymbol{x} \geq 0.$$

The feasibility, boundedness, and existence of optimal solutions, along with the relationship with its LP relaxation, are summarized in the following lemma.

**Lemma 1.** *Given $\boldsymbol{A} \in \mathbb{Q}^{m \times n}, \boldsymbol{b} \in \mathbb{Q}^m, \boldsymbol{c} \in \mathbb{Q}^m$, it holds that*

- *(I) If LP$(\boldsymbol{A}, \boldsymbol{b}, \boldsymbol{c})$ is infeasible, MILP$(\boldsymbol{A}, \boldsymbol{b}, \boldsymbol{c})$ must be infeasible.*

- *(II) If LP$(\boldsymbol{A}, \boldsymbol{b}, \boldsymbol{c})$ is feasible but unbounded, then MILP$(\boldsymbol{A}, \boldsymbol{b}, \boldsymbol{c})$ must be either infeasible or unbounded.*

- *(III) If LP$(\boldsymbol{A}, \boldsymbol{b}, \boldsymbol{c})$ is feasible and bounded, MILP$(\boldsymbol{A}, \boldsymbol{b}, \boldsymbol{c})$ might be infeasible or feasible. If we further assume MILP$(\boldsymbol{A}, \boldsymbol{b}, \boldsymbol{c})$ is feasible, it must yield an optimal solution.*

*Proof.* Conclusion (I) is trivial. Conclusion (II) is exactly (Byrd et al., 1987, Theorem 1). Conclusion (III) is a corollary of (Meyer, 1974, Theorem 2.1). To obtain (III), we first write MILP$(\boldsymbol{A}, \boldsymbol{b}, \boldsymbol{c})$ into the following form:

$$\min_{\boldsymbol{x},\boldsymbol{r}} \boldsymbol{c}^\top \boldsymbol{x} \quad \text{s.t. } \boldsymbol{A}\boldsymbol{x} + \boldsymbol{r} = \boldsymbol{b},\ \boldsymbol{x} \geq \boldsymbol{0},\ \boldsymbol{r} \geq \boldsymbol{0},\ \boldsymbol{x} \text{ is integral}$$

Then the condition (v) in (Meyer, 1974, Theorem 2.1) can be directly applied. Therefore, the feasibility and boundedness of MILP$(\boldsymbol{A}, \boldsymbol{b}, \boldsymbol{c})$ imply the existence of optimal solutions, which concludes the proof. $\square$

With Lemma 1, we could prove Theorem 1 now.

*Proof of Theorem 1.* At the beginning, we define the space of $[\boldsymbol{A}, \boldsymbol{b}, \boldsymbol{c}]$ generated based on $\mathcal{F}$ as $\mathcal{H}''$ for simplicity.

$$\mathcal{H}'' := \left\{ [\boldsymbol{A}, \boldsymbol{b}, \boldsymbol{c}] : \boldsymbol{b} = \boldsymbol{A}\boldsymbol{x} + \boldsymbol{r}, \boldsymbol{c} = \boldsymbol{A}^\top \boldsymbol{y} - \boldsymbol{s}, [\boldsymbol{A}, \boldsymbol{x}, \boldsymbol{y}, \boldsymbol{s}, \boldsymbol{r}] \in \mathcal{F} \right\}$$

Then it's enough to show that $\mathcal{H}' \subset \mathcal{H}''$ and $\mathcal{H}'' \subset \mathcal{H}'$.

We first show $\mathcal{H}'' \subset \mathcal{H}'$: For any $[\boldsymbol{A}, \boldsymbol{b}, \boldsymbol{c}] \in \mathcal{H}''$, it holds that $[\boldsymbol{A}, \boldsymbol{b}, \boldsymbol{c}] \in \mathcal{H}'$. In another word, we have to show $\mathrm{MILP}(\boldsymbol{A}, \boldsymbol{b}, \boldsymbol{c})$ to be feasible and bounded for all $[\boldsymbol{A}, \boldsymbol{b}, \boldsymbol{c}] \in \mathcal{H}''$. The feasibility can be easily verified. The boundedness can be proved by "weak duality." For the sake of completeness, we provide a detailed proof here. Define the Lagrangian as

$$\mathcal{L}(\boldsymbol{x}, \boldsymbol{y}) := \boldsymbol{c}^\top \boldsymbol{x} + \boldsymbol{y}^\top (\boldsymbol{b} - \boldsymbol{A}\boldsymbol{x})$$

Inequalities $\boldsymbol{A}\boldsymbol{x} \leq \boldsymbol{b}$ and $\boldsymbol{y} \geq \boldsymbol{0}$ imply

$$\mathcal{L}(\boldsymbol{x}, \boldsymbol{y}) \geq \boldsymbol{c}^\top \boldsymbol{x}$$

Inequalities $\boldsymbol{A}^\top \boldsymbol{y} \geq \boldsymbol{c}$ and $\boldsymbol{x} \geq \boldsymbol{0}$ imply

$$\mathcal{L}(\boldsymbol{x}, \boldsymbol{y}) \leq \boldsymbol{b}^\top \boldsymbol{y}$$

Since $\boldsymbol{x} \in \mathbb{Q}_{\geq 0}^n$ and $\boldsymbol{y} \in \mathbb{Q}_{\geq 0}^m$, it holds that

$$-\infty < \boldsymbol{c}^\top \boldsymbol{x} \leq \boldsymbol{b}^\top \boldsymbol{y} < +\infty$$

which concludes the boundedness of $\mathrm{MILP}(\boldsymbol{A}, \boldsymbol{b}, \boldsymbol{c})$.

We then show $\mathcal{H}' \subset \mathcal{H}''$: For any $\mathrm{MILP}(\boldsymbol{A}, \boldsymbol{b}, \boldsymbol{c})$ that is feasible and bounded, there must be $[\boldsymbol{A}, \boldsymbol{x}, \boldsymbol{y}, \boldsymbol{s}, \boldsymbol{r}] \in \mathcal{F}$ such that

$$\boldsymbol{b} = \boldsymbol{A}\boldsymbol{x} + \boldsymbol{r}, \tag{9}$$
$$\boldsymbol{c} = \boldsymbol{A}^\top \boldsymbol{y} - \boldsymbol{s}. \tag{10}$$

The existence of $\boldsymbol{x}, \boldsymbol{r}$, along with equation 9, is a direct conclusion of the feasibility of $\mathrm{MILP}(\boldsymbol{A}, \boldsymbol{b}, \boldsymbol{c})$. Now let's prove the existence of rational vectors $\boldsymbol{y}, \boldsymbol{s}$, along with equation 10. Since $\mathrm{MILP}(\boldsymbol{A}, \boldsymbol{b}, \boldsymbol{c})$ is feasible and bounded, according to Lemma 1, $\mathrm{LP}(\boldsymbol{A}, \boldsymbol{b}, \boldsymbol{c})$ must be feasible and bounded. Thanks to the weak duality discussed above, we conclude that $\mathrm{DualLP}(\boldsymbol{A}, \boldsymbol{b}, \boldsymbol{c})$ must be feasible and bounded. As long as $\mathrm{DualLP}(\boldsymbol{A}, \boldsymbol{b}, \boldsymbol{c})$ has an optimal solution $\boldsymbol{y}^\star$ that is rational, one can obtain equation 10 by regarding $[\boldsymbol{y}^\star, \boldsymbol{A}^\top \boldsymbol{y}^\star - \boldsymbol{c}]$ as $[\boldsymbol{y}, \boldsymbol{s}]$. Therefore, it's enough to show $\mathrm{DualLP}(\boldsymbol{A}, \boldsymbol{b}, \boldsymbol{c})$ has a rational optimal solution.

Define:

$$\boldsymbol{A}' = [\boldsymbol{A}^\top, -\boldsymbol{I}]$$
$$\boldsymbol{y}' = [\boldsymbol{y}^\top, \boldsymbol{s}^\top]^\top$$
$$\boldsymbol{b}' = [\boldsymbol{b}^\top, \boldsymbol{0}^\top]$$

Then DualLP can be written as a standard-form LP:

$$\min_{\boldsymbol{y}'} (\boldsymbol{b}')^\top \boldsymbol{y}' \quad \text{s.t. } \boldsymbol{A}' \boldsymbol{y}' = \boldsymbol{c}, \ \boldsymbol{y}' \geq \boldsymbol{0} \tag{11}$$

As long as an LP has an optimal solution, it must have a basic optimal solution Bertsimas & Tsitsiklis (1997). Specifically, we can split $\boldsymbol{A}'$ in column-based fashion as $\boldsymbol{A}' = [\boldsymbol{B}', \boldsymbol{N}']$ and split $\boldsymbol{y}'$ as $\boldsymbol{y}' = [\boldsymbol{y}_B^\top, \boldsymbol{y}_N^\top]^\top$, where $\boldsymbol{y} = \boldsymbol{0}$. Such a $\boldsymbol{y}'$ is termed a *basic optimal solution* to the LP presented in equation 11. Therefore,

$$\boldsymbol{A}' \boldsymbol{y}' = \boldsymbol{B}' \boldsymbol{y}_B + \boldsymbol{N}' \boldsymbol{y}_N = \boldsymbol{B}' \boldsymbol{y}_B = \boldsymbol{c} \implies \boldsymbol{y}_B = (\boldsymbol{B}')^{-1} \boldsymbol{c}$$

Since $\boldsymbol{B}'$ is a sub-matrix of $\boldsymbol{A}'$, $\boldsymbol{B}'$ is rational. Therefore, $(\boldsymbol{B}')^{-1}$ and $\boldsymbol{y}_B$ are rational, which implies $\boldsymbol{y}'$ is rational. This concludes the existence of rational optimal solutions of DualLP, which finishes the entire proof. $\square$

## A.2 DERIVATION OF THE LOSS FUNCTION

Here we show the derivation from the training objective in Equation. 5 towards the loss function in Equation. 6.

$$
\begin{aligned}
\log \mathbb{P}(G|G';\theta,\phi) &= \mathbb{E}_{\boldsymbol{z}\sim q_\phi(\boldsymbol{z}|G)} \log \mathbb{P}(G|G';\theta,\phi) \\
&= \mathbb{E}_{\boldsymbol{z}\sim q_\phi(\boldsymbol{z}|G)}[\log \frac{p_\theta(G|G',\boldsymbol{z})p(\boldsymbol{z})}{q_\phi(\boldsymbol{z}|G)} \frac{q_\phi(\boldsymbol{z}|G)}{p(\boldsymbol{z}|G)}] \\
&= \mathbb{E}_{\boldsymbol{z}\sim q_\phi(\boldsymbol{z}|G)} \log \frac{p_\theta(G|G',\boldsymbol{z})p(\boldsymbol{z})}{q_\phi(\boldsymbol{z}|G)} + \mathbb{E}_{\boldsymbol{z}\sim q_\phi(\boldsymbol{z}|G)}[\log \frac{q_\phi(\boldsymbol{z}|G)}{p(\boldsymbol{z}|G)}] \\
&= \mathbb{E}_{\boldsymbol{z}\sim q_\phi(\boldsymbol{z}|G)}[\log p_\theta(G|\boldsymbol{z},G')] - \mathcal{D}_{KL}[q_\phi(\boldsymbol{z}|G)||p(\boldsymbol{z})] + \mathcal{D}_{KL}[q_\phi(\boldsymbol{z}|G)||p(\boldsymbol{z}|G)] \\
&\geq \mathbb{E}_{\boldsymbol{z}\sim q_\phi(\boldsymbol{z}|G)}[\log p_\theta(G|G',\boldsymbol{z})] - \mathcal{D}_{KL}[q_\phi(\boldsymbol{z}|G)||\mathcal{N}(0,I)]
\end{aligned}
\tag{12}
$$

and thus we have

$$
\mathbb{E}_{G\sim\mathcal{G}}\mathbb{E}_{G'\sim p_{G'|G}} \log \mathbb{P}(G|G';\theta,\phi) \geq -\mathcal{L}_{\theta,\phi} \tag{13}
$$

## B SUPPLEMENTARY IMPLEMENTATION DETAILS

### B.0.1 HARDWARE, SOFTWARE AND PLATFORMS

At the hardware level, we employ an Intel Xeon Gold 6248R CPU and a Nvidia quadro RTX 6000 GPU. For tasks that exclusively run on the CPU, we utilize a single core, for tasks that run on the GPU, we set the upper limit to 10 cores. On the software side, we utilize PyTorch version 2.0.0+cu117 (Paszke et al., 2019) and PyTorch Geometric version 2.0.3 (Fey & Lenssen, 2019). We utilize PySCIPOpt solver version 3.5.0 (Maher et al., 2016a) for optimization purposes with default configurations.

### B.1 IMPLEMENTATION OF DIG-MILP

For both the encoder and the decoder, we adopt the bipartite GNN exactly the same as that in Gasse et al. (2019) as their backbones, the original codes for the backbone is publicly available[4].

**Encoder** To obtain the latent variable samples, we feed the encoder with $G$ encoded as per the method in Table. 1, we then incorporate two distinct multi-layer perceptron (MLP) layers following the backbone to output the mean and log variance of the latent variable $\boldsymbol{z}$. During the training process, we use the re-parametrization trick (Bengio et al., 2013; Maddison et al., 2016; Jang et al., 2016) to render the process of sampling $\boldsymbol{z}$ from the mean and variance differentiable. During inference, we directly sample $\boldsymbol{z} \sim \mathcal{N}(0,I)$.

**Decoder** We feed the backbone of the decoder with the incomplete graph $G'$ to obtain the latent node representations $\boldsymbol{h}^{G'} = \{\boldsymbol{h}_c^{G'}, \boldsymbol{h}_v^{G'}\}$. The backbone is then followed by seven distinct heads conditionally independent on $\boldsymbol{h}$ and $\boldsymbol{z}$, each corresponding to the prediction of: 1) the de-

Table 8: The last layer design of decoder.

| prediction | embeddings |
|---|---|
| $d, e, w, \boldsymbol{x}, \boldsymbol{r}$ | $\boldsymbol{h}_v, \boldsymbol{z}_v, v \in \mathcal{V}$ |
| $\boldsymbol{y}, \boldsymbol{s}$ | $\boldsymbol{h}_c, \boldsymbol{z}_c, c \in \mathcal{C}$ |

gree of the removed node $d_{c_i}$, 2) the edges $e(c_i, u)$ between the constraint node $c_i$ and the nodes in the other side, 3) the edge weights $w_{c_i}$, and 4) - 7) the value of $\boldsymbol{x}, \boldsymbol{y}, \boldsymbol{r}, \boldsymbol{s}$ of the new graph $\tilde{G}$. Each head is composed of layers of MLP, and takes different combinations $\boldsymbol{h}^{G'}, \boldsymbol{z}^{G'}$ as inputs, which is illustrated in Table. 8. We perform min-max normalization on all the variables to predict according to their maximum and minimum value occurred in the training dataset. Each part is modeled as a regression task, where we use the Huber Loss Huber (1992) as the criterion for each part and add them together as the total loss for decoder.

For the case of binary MILP problems, their primal, dual and slack variables could be written in the form as Equation. 14 15 16:

---

[4]https://github.com/ds4dm/learn2branch/blob/master/models/baseline/model.py

**Primal (Binary)**

$$\max_{\boldsymbol{x}} \quad \boldsymbol{c}^\top \boldsymbol{x}$$
$$\text{s.t.} \quad \mathbf{A}\boldsymbol{x} \leq \boldsymbol{b} \qquad (14)$$
$$\boldsymbol{x} \leq 1$$
$$\boldsymbol{x} \geq 0$$
$$\boldsymbol{x} \in \mathbb{Z}$$

**Dual (Binary)**
**(Linear Relaxation)**

$$\max_{y} \quad [\boldsymbol{b}^\top, 1^\top]\boldsymbol{y} \qquad (15)$$
$$\text{s.t.} \quad [\mathbf{A}^\top, I]\boldsymbol{y} \geq \boldsymbol{c}$$
$$\boldsymbol{y} \geq 0$$

**Slack (Binary)**

$$\mathbf{A}\boldsymbol{x} + \boldsymbol{r} = \boldsymbol{b}$$
$$[\mathbf{A}^\top, I]\boldsymbol{y} - \boldsymbol{s} = \boldsymbol{c} \qquad (16)$$
$$\boldsymbol{r} \geq 0$$
$$\boldsymbol{s} \geq 0$$

Considering the inherent structure of binary MILP, we can further decompose the dual solution $\boldsymbol{y}$ into two parts: $\boldsymbol{y}_1$ (corresponding to regular constraints $\mathbf{A}\boldsymbol{x} \leq \boldsymbol{b}$) and $\boldsymbol{y}_2$ (corresponding to constraints $\boldsymbol{x} \leq 1$). The encoding of binary MILP problem into a bipartite VC graph is illustrated in Table. 9. And the decoder could be models as Equation. 17.

$$p_\theta(G|G', \boldsymbol{z}) = p_\theta(d_{c_i}|\boldsymbol{h}_{c_i}^{G'}, \boldsymbol{z}_{c_i}) \cdot \prod_{u \in \mathcal{V}} p_\theta(e(c_i, u)|\boldsymbol{h}_\mathcal{V}^{G'}, \boldsymbol{z}_\mathcal{V}) \cdot \prod_{u \in \mathcal{V}: e(c_i, u) = 1} p_\theta(w_{c_i}|\boldsymbol{h}_\mathcal{V}^{G'}, \boldsymbol{z}_\mathcal{V})$$
$$\cdot \prod_{u \in \mathcal{C}} p_\theta(\boldsymbol{y}_{1u}|\boldsymbol{h}_\mathcal{C}^{G'}, \boldsymbol{z}_\mathcal{C}) p_\theta(\boldsymbol{r}_u|\boldsymbol{h}_\mathcal{C}^{G'}, \boldsymbol{z}_\mathcal{C}) \cdot \prod_{u \in \mathcal{V}} p_\theta(\boldsymbol{x}_u|\boldsymbol{h}_\mathcal{V}^{G'}, \boldsymbol{z}_\mathcal{V}) p_\theta(\boldsymbol{s}_u|\boldsymbol{h}_\mathcal{V}^{G'}, \boldsymbol{z}_\mathcal{V}) p_\theta(\boldsymbol{y}_{2u}|\boldsymbol{h}_\mathcal{V}^{G'}, \boldsymbol{z}_\mathcal{V}), \qquad (17)$$

where the decoder of DIG-MILP specifically designed for binary MILP partitions the predicted dual solution $\boldsymbol{y}$ into two segments $\boldsymbol{y}_1, \boldsymbol{y}_2$ and predict each segment separately.

Table 9: V-C encoding for binary MILP.

| object | feature |
|---|---|
| constraint | all 0's |
| node | $\boldsymbol{y}_1 = \{\boldsymbol{y}_{11}...\boldsymbol{y}_{1m}\}$ |
| $\mathcal{C} = \{c_1...c_m\}$ | $\boldsymbol{r} = \{\boldsymbol{s}_1...\boldsymbol{s}_m\}$ |
| variable | all 1's |
| node | $\boldsymbol{x} = \{\boldsymbol{x}_1...\boldsymbol{x}_n\}$ |
| | $\boldsymbol{s} = \{\boldsymbol{r}_1...\boldsymbol{r}_n\}$ |
| $\mathcal{V} = \{v_1...v_n\}$ | $\boldsymbol{y}_2 = \{\boldsymbol{y}_{21}...\boldsymbol{y}_{2n}\}$ |
| edge $\mathcal{E}$ | non-zero weights in $\boldsymbol{A}$ |

**Hyper-parameters** Across the four datasets, we set the same learning rate for DIG-MILP as $1e-3$. We use the Adam optimizer (Kingma & Ba, 2014). For the SC, we set the $\alpha$ in $\mathcal{L}_{\theta,\phi}$ as 5, for the CA, the CVS, and the IIS, we set $\alpha$ as 150. We use the random seed as 123 for DIG-MILP training across all the four datasets.

### B.2 IMPLEMENTATION OF BASELINE

**'Bowly'** Here we show the implementation of generating instances from scratch with the baseline Bowly (Bowly, 2019). The generation of matrix $\mathbf{A}$ is illustrated in the algorithm. 3. With the generated adjacency matrix $\mathbf{A}$, where we manipulate the hyper-parameters during the generation process to ensure that the statistical properties of $\mathbf{A}$ align as closely as possible with the original dataset. Specifically, we keep the size of graph $(m, n)$ the same as the original dataset and uniformly sample $p_v, p_c$ from $[0, 1]$ for all the four datasets. For the other hyper-parameter settings, see Tables. 10.

Table 10: The hyper-parameter selection of the Bowly baseline.

| | density | $\mu_\mathbf{A}$ | $\sigma_\mathbf{A}$ |
|---|---|---|---|
| SC | $\mathcal{U}\{0.15, 0.20, 0.25, 0.30, 0.35\}$ | -1 | 0 |
| CA | 0.05 | 1 | $\mathcal{U}(0.1, 0.3)$ |
| CVS | 0.0013 | 0.2739 | 0.961 |
| IIS | 0.0488 | -1 | 0 |

Then we uniformly sample the solution space $x, y, s, r$ with intervals defined by their corresponding maximum and minimum from the training dataset. Then we deduce $b, c$ to get the new MILP instances.

---

**Algorithm 3** Bowly - generation of matrix $\mathbf{A}$

---

**Require:** $n \in [1, \infty), m \in [1, \infty), \rho \in (0, 1], p_v \in [0, 1], p_c \in [0, 1], \mu_A \in (-\infty, \infty), \sigma_A \in (0, \infty)$

**Ensure:** Constraint matrix $\mathbf{A} \in \mathbb{Q}^{m \times n}$

1: Set target variable degree $d_{(u_i)} = 1$ for randomly selected i,0 for all others
2: Set target constraint degree $d_{(u_i)} = 1$ for randomly selected i,0 for all others
3: $e \leftarrow 1$
4: **while** $e < \rho m n$ **do**
5:     $s \leftarrow$ draw $n$ values from $U(0, 1)$
6:     $t \leftarrow$ draw $m$ values from $U(0, 1)$
7:     Increment the degree of variable node $i$ with maximum $p_v \frac{d(u_i)}{e} + s_i$
8:     Increment the degree of constraint node $j$ with maximum $p_c \frac{d(v_j)}{e} + t_j$
9:     $e \leftarrow e + 1$
10: **end while**
11: **for** $i = 1, ..., n$ **do**
12:     **for** $j = 1, ..., m$ **do**
13:         $r \leftarrow$ draw from $U(0, 1)$
14:         **if** $r < \frac{d(u_i)d(v_j)}{e}$ **then**
15:             Add edge $(i, j)$ to VC
16:         **end if**
17:     **end for**
18: **end for**
19: **while** $\min((d(u_i), d(v_j)) = 0$ **do**
20:     Choose $i$ from $\{i | d(u_i) = 0\}$, or randomly if all $d(u_i) > 0$
21:     Choose $j$ from $\{j | d(v_j) = 0\}$, or randomly if all $d(v_j) > 0$
22:     Add edge $(i, j)$ to VC
23: **end while**
24: **for** $(i, j) \in E(VC)$ **do**
25:     $a_{ij} = \mathcal{N}(\mu_\mathbf{A}, \sigma_\mathbf{A})$
26: **end for**
27: **return A**

---

**'Random'** We use exactly the same network architecture and generation process as DIG-MILP. The key difference is that instead of utilizing the trained NN, we uniformly sample the variables $d_{c_i}, e(c_i, u), w_{c_i}, y_1, s, x, r, y_2$ required for decoder prediction within intervals delineated by the maximum and minimum values of each variable from the training set, simulating the random parameters of an untrained neural network.

### B.3 IMPLEMENTATION OF THE STRUCTURAL STATISTICAL CHARACTERISTICS

The explanation of various statistical metrics used for comparing the structural similarity of MILP problem instances is detailed as shown in Table. 11. Specific numerical values for different metrics for the SC and CA problems can be found in Table. 13 and Table. 14, respectively.

For each statistic metric $i$ shown in Table. 11, we begin by collecting lists of the values from four data sources: the original dataset, the data generated by the 'Bowly' baseline, the data generated by the 'random' baseline, and data generated by DIG-MILP. Each data source contains 1000 instances. We then employ the lists from the four data sources to approximate four categorical distributions. Utilizing the *numpy.histogram* function, we set the number of bins to the default value of 10, with the min and max values derived from the collective minimum and maximum of a given metric across the four data sources, respectively. Next, we employ Jensen-Shannon (JS) divergence $D_{js}^i$ via the function *scipy.spatial.distance.jensenshannon* (Virtanen et al., 2020) to quantify the divergence between the original samples and the rest three data sources, resulting in score$_i$ for each statistical metric.

Table 11: Explanation of the statistic metrics of the MILP instances

| name | explanation |
|---|---|
| density mean | the average number of non zero values in the constraint matrix |
| cons degree mean | the average number of constraint node degree |
| cons degree std | the standard variance of constraint node degree |
| var degree mean | the average number of variable node degree |
| var degree std | the standard variance of variable node degree |
| $b$ mean | the average $b$ value |
| $b$ std | the standard variance of $b$ value |
| $c$ mean | the average value of $c$ |
| $c$ std | the standard variance of $c$ value |

$$\text{score}_i = (\max(D_{js}) - D_{js}^i)/(\max(D_{js}) - \min(D_{js})), \tag{18}$$

where $\max(D_{js}), \min(D_{js})$ are the maximum and minimum of JS divergence across all the metrics.

Then we average the score for each statistic metric to obtain the final similarity score, as is shown in Table. 3:

$$\text{score} = \frac{1}{9}\sum_{i=1}^{9}\text{score}_i. \tag{19}$$

## B.4   IMPLEMENTATION OF DATA SHARING FOR SOLVER CONFIGURATION TUNING

Below are the hyper-parameters that we randomly sample to test the positive-correlation of different dataset pairs. We adhere to the configuration established in mainstream solver tuning literature to select the parameters requiring adjustment Hutter et al. (2011); Lindauer & Hutter (2018); Lindauer et al. (2022), . For a detailed explanation of each parameter, please refer to the SCIP documentation[5].

Table 12: The selected SCIP hyper-parameters and the range to randomly select from.

| params | whole range/choice | default | our range/choice |
|---|---|---|---|
| branching/scorefunc | s, p, q | s | s, p, q |
| branching/scorefac | [0, 1] | 0.167 | [0, 1] |
| branching/preferbinary | True, False | False | True, False |
| branching/clamp | [0,0.5] | 0.2 | [0,0.5] |
| branching/midpull | [0,1] | 0.75 | [0,1] |
| branching/midpullreldomtrig | [0,1] | 0.5 | [0,1] |
| branching/lpgainnormalize | d, l, s | s | d, l, s |
| lp/pricing | l, a, f, p, s, q, d | l | l, a, f, p, s, q, d |
| lp/colagelimit | [-1,2147483647] | 10 | [0,100] |
| lp/rowagelimit | [-1,2147483647] | 10 | [0,100] |
| nodeselection/childsel | d, u, p, I, l, r, h | h | d, u, p, I, l, r, h |
| separating/minortho | [0,1] | 0.9 | [0,1] |
| separating/minorthoroot | [0,1] | 0.9 | [0,1] |
| separating/maxcuts | [0,2147483647] | 100 | [0,1000] |
| separating/maxcutsroot | [0,2147483647] | 2000 | [0,10000] |
| separating/cutagelimit | [-1,2147483647] | 80 | [0,200] |
| separating/poolfreq | [-1,65534] | 10 | [0,100] |

## B.5   IMPLEMENTATION OF OPTIMAL VALUE PREDICTION VIA ML

**Neural Network Architecture** In this downstream task, We also use the bipartite GNN backbone which is exactly the same as that in Gasse et al. (2019). We use an MLP layer and global mean pooling to produce the optimal objective value prediction. The learning rate is set as $1e-3$.

---

[5]https://www.scipopt.org/doc/html/PARAMETERS.php

Table 13: Statistic value comparison across the original dataset and the generated datasets with different constraints replacement rates on the set covering (SC) problem. 'resolving time' calculates under default configuration of pySCIPopt. 'density' represents the ratio of non zero entries in the constraint matrix. 'cons degree' denotes the degree of constraint nodes, 'var degree' stands for the degree of variable nodes. $b$ denotes the right hand side vector of the MILP, and $c$ is the objective coefficient vector.

|  | replace ratio | resolving time (s) | density mean | cons degree mean | cons degree std | var degree mean | var degree std | b mean | b std | c mean | c std |
|---|---|---|---|---|---|---|---|---|---|---|---|
| original | - | 0.821 | 0.251 | 100.700 | 8.447 | 50.350 | 6.854 | -1.0 | 0.0 | 50.490 | 28.814 |
| Bowly | - |  | 0.205 | 82.312 | 35.131 | 41.305 | 21.628 | 1.484 | 3.504 | 403.208 | 198.571 |
| random | 0.01 | 127.723 | 0.251 | 100.774 | 9.853 | 50.387 | 6.841 | 1.294 | 3.045 | 422.65 | 65.078 |
| random | 0.05 | 143.883 | 0.253 | 101.039 | 14.070 | 50.519 | 6.787 | 1.218 | 3.123 | 431.422 | 66.082 |
| random | 0.10 | 187.851 | 0.253 | 101.357 | 17.706 | 50.678 | 6.727 | 1.164 | 3.210 | 441.696 | 67.250 |
| random | 0.20 | 304.216 | 0.255 | 101.900 | 22.808 | 50.950 | 6.607 | 1.062 | 3.351 | 460.696 | 69.379 |
| random | 0.50 | 1312.595 | 0.258 | 103.348 | 31.305 | 51.674 | 6.375 | 0.664 | 3.629 | 509.337 | 74.864 |
| ours | 0.01 | 83.681 | 0.251 | 100.700 | 8.876 | 50.350 | 7.431 | -0.515 | 1.351 | 44.863 | 0.939 |
| ours | 0.05 | 70.476 | 0.251 | 100.712 | 10.202 | 50.356 | 9.977 | -0.456 | 1.386 | 44.958 | 0.984 |
| ours | 0.10 | 54.650 | 0.251 | 100.738 | 11.365 | 50.369 | 13.354 | -0.413 | 1.441 | 45.057 | 1.032 |
| ours | 0.20 | 54.830 | 0.251 | 100.754 | 12.872 | 50.377 | 19.992 | -0.368 | 1.576 | 45.112 | 1.071 |
| ours | 0.50 | 22.462 | 0.252 | 100.830 | 14.433 | 50.415 | 37.017 | -0.005 | 1.271 | 44.967 | 1.872 |

Table 14: Statistic value comparison across the original dataset and the generated datasets with different constraints replacement rates on the combinatorial auction (CA) problem. 'resolving time' calculates under default configuration of pySCIPopt. 'density' represents the ratio of non zero entries in the constraint matrix. 'cons degree' denotes the degree of constraint nodes, 'var degree' stands for the degree of variable nodes. $b$ denotes the right hand side vector of the MILP, and $c$ is the objective coefficient vector.

|  | replace ratio | resolving time (s) | density mean | cons degree mean | cons degree std | var degree mean | var degree std | b mean | b std | c mean | c std |
|---|---|---|---|---|---|---|---|---|---|---|---|
| original | - | 1.360 | 0.050 | 14.538 | 13.834 | 5.578 | 3.253 | 1.0 | 0.0 | 330.999 | 234.444 |
| Bowly | - | 0.281 | 0.048 | 14.415 | 13.633 | 5.544 | 7.262 | 1.668 | 1.617 | 510.211 | 1101.065 |
| random | 0.01 | 0.416 | 0.051 | 14.664 | 13.970 | 5.634 | 3.240 | 1.748 | 1.602 | 524.961 | 563.436 |
| random | 0.05 | 0.502 | 0.054 | 15.225 | 14.531 | 5.878 | 3.201 | 1.792 | 1.647 | 560.369 | 561.074 |
| random | 0.10 | 0.555 | 0.056 | 15.877 | 15.088 | 6.152 | 3.161 | 1.855 | 1.706 | 598.047 | 555.956 |
| random | 0.20 | 0.821 | 0.061 | 17.098 | 15.953 | 6.658 | 3.106 | 1.966 | 1.797 | 669.168 | 552.853 |
| random | 0.30 | 1.056 | 0.065 | 18.186 | 16.527 | 7.105 | 3.070 | 2.053 | 1.850 | 735.284 | 548.606 |
| random | 0.50 | 2.353 | 0.072 | 19.959 | 17.222 | 7.837 | 3.006 | 2.267 | 1.972 | 841.971 | 545.471 |
| ours | 0.01 | 0.361 | 0.050 | 14.490 | 13.776 | 5.565 | 3.253 | 1.645 | 1.348 | 361.711 | 264.798 |
| ours | 0.05 | 0.360 | 0.050 | 14.361 | 13.609 | 5.535 | 3.286 | 1.609 | 1.325 | 351.417 | 261.927 |
| ours | 0.10 | 0.301 | 0.050 | 14.205 | 13.401 | 5.500 | 3.366 | 1.589 | 1.329 | 342.702 | 261.313 |
| ours | 0.20 | 0.217 | 0.049 | 13.819 | 12.854 | 5.412 | 3.586 | 1.525 | 1.315 | 324.282 | 260.848 |
| ours | 0.30 | 0.140 | 0.047 | 13.454 | 12.330 | 5.344 | 3.847 | 1.454 | 1.280 | 304.911 | 260.949 |
| ours | 0.50 | 0.055 | 0.045 | 12.869 | 11.379 | 5.254 | 4.282 | 1.350 | 1.233 | 271.474 | 255.515 |

# C SUPPLEMENTARY EXPERIMENT RESULTS

## C.1 STATISTICAL CHARACTERISTICS OF THE GENERATED INSTANCES

We show the specific value of each statistic metric of the original dataset, and the datasets generated by the baselines as well as DIG-MILP on the SC and the CA problem in Table. 13 and Table. 14 respectively.

## C.2 DATA SHARING FOR SOLVER CONFIGURATION TUNING

**CVS and IIS** There are five total instances in CVS, comprising three for training DIG-MILP and the downstream predictor and two for testing. The IIS has two instances, one for training and one for testing (with allocation based on alphabetical order). Please refer to Table. 15 for the model's performance. 'ground truth' corresponds to the true values of the optimal objectives for each problem. Models trained exclusively on the 'original' training set exhibit superior fitting and more accurate predictions on the training set itself. However, models trained on the datasets where we introduce 20 additional newly generated instances by DIG-MILP with varying constraint replacement ratio $\gamma$ not only demonstrate minimal gap in prediction on the training set towards the models trained solely on the original data compared with the baselines, but also showcase improved predictive performance on previously unseen test sets. This underscores the notion that the DIG-MILP-generated data can indeed increase structural and solution label diversity to a certain extent, thereby enhancing the generalization capability and overall performance of the models. Again, similar to the previous

Table 15: The predicted value and relative mean square error (MSE) of the optimal objective value on the CVS and the IIS problem. In the CVS, 'cvs08r139-94','cvs16r70-62','cvs16r89-60' are used as training data, 'cvs16r106-72','cvs16r128-89' are used as testing data. In the IIS, 'iis-glass-cov' is used as the training data, 'iis-hc-cov' is used as the testing data. 'original' shows the performance of the model trained merely on the three (CVS) or single (IIS) original training instances.

| | | in-distribution | | | | | | out-of-distributio | | | | in-distribution | | out-of-distribution | |
| | | cvs08r139-94 | | cvs16r70-62 | | cvs16r89-60 | | **cvs16r106-72** | | **cvs16r128-89** | | iis-glass-cov | | **iis-hc-cov** | |
| dataset | ratio | value | msre | value | msre | value | msre | value | msre | value | msre | value | msre | value | msre |
| --- | --- | --- | --- | --- | --- | --- | --- | --- | --- | --- | --- | --- | --- | --- | --- |
| ground truth | - | 116 | 0 | 42 | 0 | 65 | 0 | 81 | 0 | 97 | 0 | -17 | 0 | -21 | 0 |
| original | - | **115.994** | **2e-9** | **41.998** | **1e-9** | **64.997** | **1e-9** | 77.494 | 0.001 | 89.258 | 0.006 | **-20.999** | **3e-10** | -94.451 | 20.756 |
| Bowly | - | 65.712 | 0.187 | 82.353 | 0.923 | 66.858 | 8e-6 | 61.504 | 0.057 | 66.045 | 0.101 | -88.756 | 17.816 | -88.756 | 17.816 |
| random | 0.01 | 138.459 | 0.037 | 45.312 | 0.006 | 67.875 | 0.001 | 58.754 | 0.075 | 68.192 | 0.088 | -22.263 | 3e-4 | -83.146 | 15.139 |
| random | 0.05 | 163.412 | 0.167 | 34.571 | 0.031 | 45.605 | 0.089 | 41.110 | 0.242 | 24.952 | 0.551 | -20.695 | 2e-4 | -82.297 | 14.753 |
| random | 0.10 | 116.824 | 5e-5 | 60.440 | 0.192 | 79.152 | 0.047 | 68.641 | 0.023 | 79.321 | 0.033 | -20.991 | 1e-7 | -807.680 | 2163.238 |
| random | 0.20 | 144.962 | 0.062 | 79.849 | 0.812 | 99.552 | 0.282 | 71.821 | 0.0128 | 99.898 | 8e-4 | -21.678 | 0.001 | -227.610 | 153.482 |
| random | 0.50 | 159.807 | 0.142 | 49.364 | 0.030 | 65.213 | 1e-5 | 103.960 | 0.080 | 122.321 | 0.068 | -21.633 | 9e-3 | -100.224 | 23.966 |
| DIG-MILP | 0.01 | 116.981 | 7e-5 | 42.197 | 2e-5 | 64.876 | 3e-6 | 78.646 | 8e-4 | **96.831** | **3e-6** | -20.933 | 1e-5 | -90.556 | 18.721 |
| DIG-MILP | 0.05 | 161.558 | 0.154 | 26.181 | 0.141 | 23.439 | 0.408 | 66.119 | 0.033 | 76.119 | 0.046 | -21.108 | 2e-5 | -61.217 | 6.765 |
| DIG-MILP | 0.10 | 118.609 | 5e-4 | 45.461 | 0.006 | 67.216 | 0.001 | **80.706** | **1e-5** | 95.745 | 1e-4 | -20.976 | 1e-6 | -65.385 | 8.101 |
| DIG-MILP | 0.20 | 114.622 | 1e-4 | 42.933 | 4e-4 | 62.627 | 0.001 | 83.379 | 8e-4 | 120.641 | 0.0594 | -20.159 | 0.001 | **-55.926** | **5.243** |
| DIG-MILP | 0.50 | 120.361 | 0.001 | 44.472 | 0.003 | 69.287 | 0.004 | 84.870 | 0.002 | 104.333 | 0.005 | -21.009 | 2e-7 | -90.427 | 18.655 |

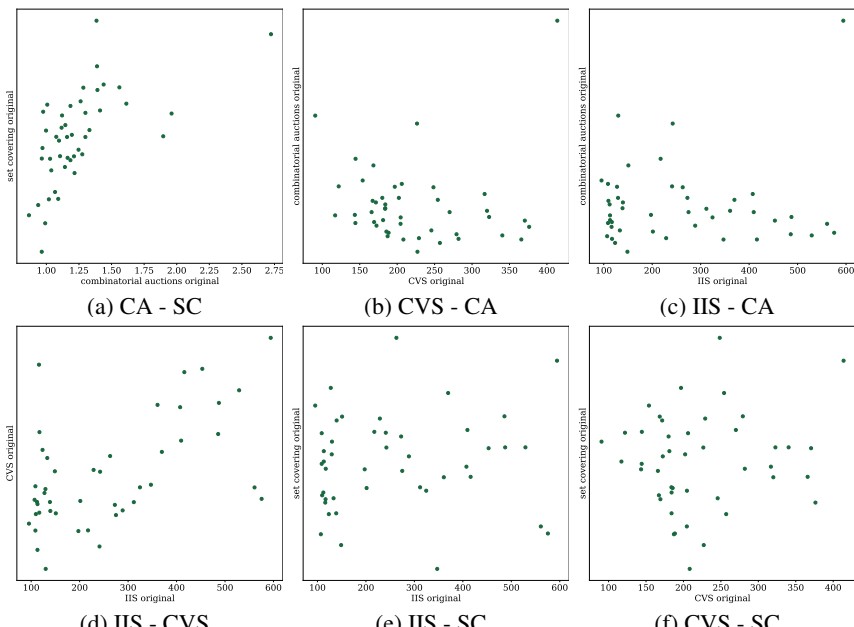

(a) CA - SC      (b) CVS - CA      (c) IIS - CA

(d) IIS - CVS      (e) IIS - SC      (f) CVS - SC

Figure 4: The solution time of SCIP with different parameter sets across different original datasets.

two experiments, 'Bowly' degrades the predictive performance of the model, 'random' results in marginal improvement in out-of-distribution prediction accuracy.

We present the visual results for CA, SC, and IIS datasets, see Fig. 5, 6, 7.

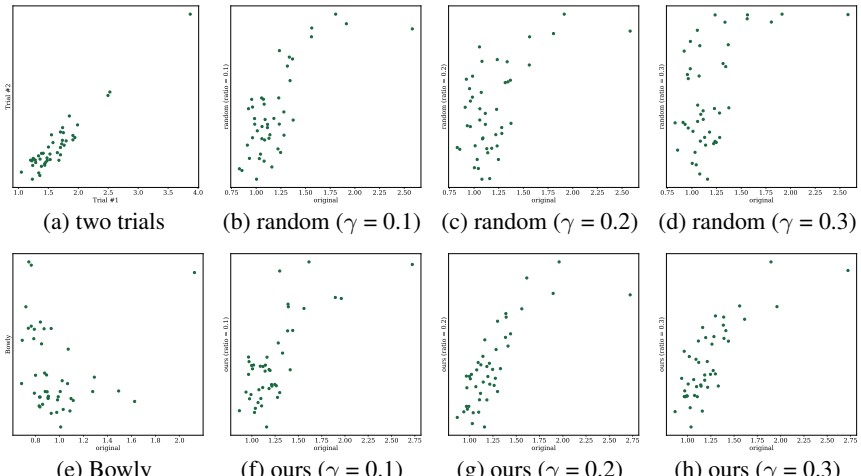

Figure 5: The solution time of SCIP on the CA with 45 different hyper-parameter sets.

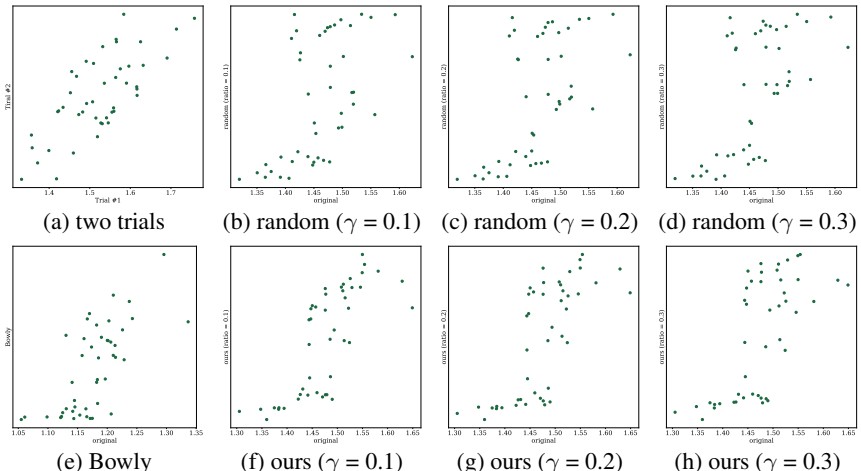

Figure 6: The solution time of SCIP on the SC with 45 different hyper-parameter sets.

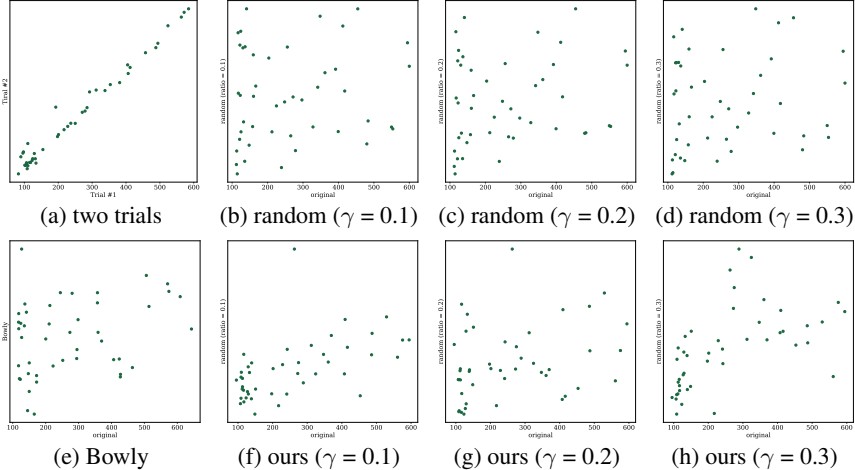

Figure 7: The solution time of SCIP on the IIS with 45 different hyper-parameter sets.

