# OpenReview forum: "DIG-MILP: a Deep Instance Generator for Mixed-Integer Linear Programming with Feasibility Guarantee"
_ICLR.cc/2024/Conference — Submitted to ICLR 2024_

### Official Review · Reviewer_HSpe · 2023-10-24

**Soundness:** 3 good
**Presentation:** 3 good
**Contribution:** 2 fair
**Rating:** 3
**Confidence:** 4

**Summary:**

In this paper, the authors propose a MILP instance generator with the help of the variational auto-encoder (VAE), aiming at generating enough MILP instances for academic research and industry usage. The authors first prove the boundedness and feasibility when generating instances using the dual formation, which ensures the correctness of the proposed generator DIG-MILP. In the training step, DIG-MILP continues to mask a random node and the connections to it, and use the incomplete instance as input to the VAE, aiming to reconstruct the instance. This task is suitable for VAE and helps the model to learn how to generate more MILP instances. The proposed method is well-designed and the proofs are sufficient. The experiments on multiple datasets and scenarios show the performance of the proposed method.

**Strengths:**

1. There is indeed a need to generate more high-quality MILP instances in both the industry and academia. Therefore, this work is necessary and interesting to the community.
2. The proposed VAE framework for generating MILP instances is self-contained and reasonable, as VAE is demonstrated to be useful in many instance generation tasks.
3. The authors have proven the generated instances to be feasible-bounded.

**Weaknesses:**

1. In the pipeline of the proposed method, I see that one node will be removed and the origin G is changed to G'. I wonder if this one-node change is too small to identify. In common MILP datasets, the size of nodes/constraints is more than thousands, and for the industry area, the size is even larger.
2. In section 3.2, the authors provide the feasibility guarantee for the proposed methods. But I am curious about unfeasible instances, can the proposed framework generate unfeasible MIP instances? In my view, it is okay when a MIP instance is infeasible and this situation is common in the real world.
3. The metrics in the experiments are confusing to me. The authors mentioned: "The similarity score is derived from the Jensen-Shannon (JS) divergence (the lower the better) between each metric of the generated and original data", while in the caption of table 1 said: "The similarity score ↑ between the original and generated data." The similarity score is lower the better or higher the better? Actually, I wonder what we expect about the similarity score. I mean, more similarity could increase Authenticity while less similarity could increase Diversity. Both cases seem to mean something.
4. About the datasets in the experiments, I see that for small datasets SC and CA the amount of instances is 1000, but for large datasets CVS and IIS the amount is less than 10. This gap looks very strange to me. Moreover, in, table 2, does training and testing on merely 3 instances lead to an overfitting to these instances? I think the datasets CVS and IIS need to be further refined. I understand that they are selected from MIPLIB17, but MIPLIB17 at least has hundreds of instances. If the MIPLIB17 is too hard, I think NeurIPS 2021 ML4CO[1] datasets could be more suitable.


[1] Maxime Gasse, Simon Bowly, Quentin Cappart, Jonas Charfreitag, Laurent Charlin, Didier Ch´etelat, Antonia Chmiela, Justin Dumouchelle, Ambros Gleixner, Aleksandr M Kazachkov, et al. The machine learning for combinatorial optimization competition (ml4co): Results and insights. In NeurIPS 2021 Competitions and Demonstrations Track, pp. 220–231. PMLR, 2022.

**Questions:**

Please refer to the weakness part of my review. Admittedly, I am not familiar with generative models, so my questions are mainly about this part. However, based on my experience, I think the dataset problem is more severe, as I do not think training and testing on only 3 instances is suitable for meeting the bar of ICLR.

Besides, some closely related works on graph generation especially SAT instance generation are missed, including [1], [2], [3], all of which exhibit highly similar task structures for generating bipartite combinatorial optimization problems.

If there are some fatal mistakes in my review, please point it to me.


[1] G2SAT: Learning to Generate SAT Formulas. NeurIPS 2019.

[2] On the Performance of Deep Generative Models of Realistic SAT Instances. SAT 2022.

[3] HardSATGEN: Understanding the Difficulty of Hard SAT Formula Generation and A Strong Structure-Hardness-Aware Baseline. SIGKDD 2023.

---

> ### Author Response · Authors · 2023-11-22
> **Response to reviewer HSpe**
>
> Thank you, Reviewer HSpe, for your valuable suggestions. We address your queries as follows:
>
> 1. **Changing One Node at a Time**:
>    We appreciate your insightful suggestion. DIG-MILP currently supports iteratively selecting and altering a single node in the graph. However, we acknowledge that selecting and changing multiple nodes simultaneously might be an alternative and potentially effective approach, and we would leave this as a future work.
>
> 2. **Feasible vs. Infeasible Instances**:
>    As mentioned in our universal response, we argue that given that almost all the existing scalable MILP datasets are feasible, it is naturally more interesting to generate feasible-bounded MILP instances. In real-world industrial applications, MILP problems modeled from real-world applications are usually feasible, or infeasible problems are transformed into feasible ones for solver application. Therefore, we argue that training models on feasible datasets to generate feasible MILP instances could be a justified approach.
>
> 3. **JS-Divergence and Similarity Score**:
> We use JS-divergence where a smaller value indicates higher similarity of two distributions. The similarity score is calculated in the way like (1 - JS divergence), implying that a larger score is preferable. Despite aiming for diversity in new data, it's crucial that the new MILP instances represent problem types similar to the original. This relevance ensures their practical utility in tasks like solver tuning and as augmentation in ML training. We measure such similarity to the original problem distribution by comparing various statistical metrics on graphs, leading to our comprehensive scoring metric that incorporates these diverse measures.
>
> 4. **Training Data for CVS and IIS**:
> The instances in MIPLIB2017 are of thousands of constraints and variables. In each training iteration, we sample a training batch by randomly choosing a node and delete it from the original bipartite graph, which leads to new training examples during the training process. Also for the downstream task, we train the model with the three original data with 20 newly generated instances by our model. We may consider larger datasets for further experiments.
>
> We hope these responses clarify your concerns.

---

### Official Review · Reviewer_T4Ma · 2023-10-29

**Soundness:** 2 fair
**Presentation:** 3 good
**Contribution:** 2 fair
**Rating:** 3
**Confidence:** 5

**Summary:**

The manuscript presents DIG-MILP, an innovative deep generative approach tailored for Mixed Integer Linear Programming (MILP) generation. Unlike traditional MILP generation methods, DIG-MILP eschews the need for domain-specific insights. A significant attribute of DIG-MILP is its assurance of the feasibility and boundedness of the crafted data. The generative spectrum of DIG-MILP spans all feasible and bounded MILPs, endowing it with the prowess to produce a "variety" of instances. Experimental analyses underscore DIG-MILP's promise in: (S1) facilitating MILP data dissemination for solver hyperparameter optimization without publishing original datasets, and (S2) data enrichment to bolster the robustness of machine learning techniques dedicated to tackling MILPs.

**Strengths:**

1.	It is impressive that the authors propose to use VAE sampling in dual space to generate MILP instance. In this way, the feasibility guarantee is directly obtained.
2.	The experimental results endorse that the proposed techniques indeed facilitate the enhancement of MILP solver and business scenarios.

**Weaknesses:**

1.	It seems that the literature review lacks of the most related paper that recently published in NeurIPS 2023 (Geng, Zijie, Xijun Li, Jie Wang, Xiao Li, Yongdong Zhang, and Feng Wu. "A Deep Instance Generative Framework for MILP Solvers Under Limited Data Availability." arXiv preprint arXiv:2310.02807 (2023).) This submission is highly similar to the above paper. Thus the authors are supposed to highlight the largest difference and improvement from the mentioned paper.
2.	Similar to the above point, the experimental setting of this paper highly resembles one in the paper “A Deep Instance Generative Framework for MILP Solvers Under Limited Data Availability”. Thus, Are the authors supposed to compared their proposed method with one proposed in that paper?
3.	It is hard to read several Figures. Because the font size of axis in those figures is too small

**Questions:**

1.	Please clarify more in-depth about the feature mentioned in Table 1, especially the all 0’s of y, r and the all 1’s about x, s.
2.	It is not that dogmatical to claim that the boundness and feasibility of the generated instances can sure the authenticity of the produced data. The authenticity of MILP dataset can be defined in kinds of perspectives. Can you give more evidences to support your claim?

---

> ### Author Response · Authors · 2023-11-22
> **Response to reviewer T4Ma**
>
> Thank you, Reviewer T4Ma, for your insightful questions. We address them as follows:
> 1. **Difference and Improvement Over Paper [1]**:
> As stated in our universal response, our work fundamentally differs from and is concurrent with [1]. We encourage you to refer to our universal response for a detailed explanation.
> 2. **Clarification on Features in Table 1**:
> The all 1’s and all 0’s denote the feature encoding in the first dimension of the node input feature. We use 1 and 0 to classify the variable node and constraint nodes from the bipartite graph.
> 3. **Authenticity of Generated MILP Instances**:
> We assert that feasible-bounded MILP instances are 'authentic’ because 1)most existing MILP datasets for training are feasible, and 2) MILP problems in industrial applications are typically feasible. Infeasible problems are generally transformed into feasible ones before being solved.
> Therefore, we believe training models on feasible datasets to generate feasible MILP instances is appropriate. Please refer to the universal response for more detailed explanation.
> 4. **Legibility of Figures**:
> Thank you for pointing out the issue with the font size in our figures. We will increase the font size to enhance readability in the revised manuscript.
> We appreciate your feedback and hope these responses clarify your concerns.
>
> [1] Geng, Zijie, et al. "A Deep Instance Generative Framework for MILP Solvers Under Limited Data Availability." Neural Information Processing Systems (2023).

---

### Official Review · Reviewer_KTFt · 2023-10-30

**Soundness:** 2 fair
**Presentation:** 3 good
**Contribution:** 2 fair
**Rating:** 3
**Confidence:** 4

**Summary:**

This paper studies the problem of instance generation for mixed-integer linear programming (MILP). The authors propose a deep generative framework based on variational auto-encoder (VAE) to capture complex structural characteristics from limited MILP data and generate instances that resemble the original data. Experiments demonstrate the proposed method outperforms baselines on various benchmarks.

**Strengths:**

1.	This paper proposes to leverage the MILP duality theory to ensure the boundedness and feasibility of the generated instances.
2.	Experiments demonstrate the proposed method outperforms baselines on various benchmarks.

**Weaknesses:**

1.	The technical novelty of the proposed method is incremental, as the proposed method primarily use the existing VAE [1, 2] model to generate MILP instances. The authors may want to explain the technical novelty of the proposed methods in detail.
2.	The relationship between the theoretical derivations (i.e., Theorem 1) and the proposed MILP generation pipeline based on the VAE [1, 2] model is unclear.
3.	The motivation of using the VAE [1, 2] model to generate MILP instances is unclear. The popular generative models include VAE [1, 2], Generative Adversarial Network (GAN) [3], and diffusion model [4]. The authors may want to explain the motivation of using the VAE [1, 2] model rather than the other generative models.
4.	I found the proposed method is similar to one recent work at NIPS [5]. The authors may want to explain the differences between their proposed method and the recent work [5] in detail.
5.	The experiments are insufficient. First, it would be more convincing if the authors could evaluate their method on large-scale benchmarks, such as instances from the MIPLIB with over 100,000 variables and 100,000 constraints. Second, the authors may want to evaluate their method on popular downstream tasks, such as learning to cut [6, 7] and learning to branch [8, 9]. Third, the baselines are insufficient. The authors may want to compare their method to G2SAT [10], which is the first deep generative framework that learns to generate Boolean Satisfiability (SAT) problems.

Overall, I would lean toward rejection due to the aforementioned concerns, while I would raise my score if the authors could properly address these concerns.

[1] Kingma, Diederik P., and Max Welling. "Auto-encoding variational bayes." The Second International Conference on Learning Representations, 2014.

[2] Kipf, Thomas N., and Max Welling. "Variational graph auto-encoders." arXiv preprint arXiv:1611.07308 (2016).

[3] Goodfellow, Ian, et al. "Generative adversarial nets." Advances in neural information processing systems 27 (2014).

[4] Ho, Jonathan, Ajay Jain, and Pieter Abbeel. "Denoising diffusion probabilistic models." Advances in neural information processing systems 33 (2020): 6840-6851.

[5] Geng, Zijie, et al. "A Deep Instance Generative Framework for MILP Solvers Under Limited Data Availability." Neural Information Processing Systems (2023).

[6] Paulus, Max B., et al. "Learning to cut by looking ahead: Cutting plane selection via imitation learning." International conference on machine learning. PMLR, 2022.

[7] Wang, Zhihai, et al. "Learning Cut Selection for Mixed-Integer Linear Programming via Hierarchical Sequence Model." The Eleventh International Conference on Learning Representations. 2023.

[8] Gupta, Prateek, et al. "Hybrid models for learning to branch." Advances in neural information processing systems 33 (2020): 18087-18097.

[9] Gasse, Maxime, et al. "Exact combinatorial optimization with graph convolutional neural networks." Advances in neural information processing systems 32 (2019).

[10] You, Jiaxuan, et al. "G2SAT: Learning to generate SAT formulas." Advances in neural information processing systems 32 (2019).

**Questions:**

1.	What is the technical novelty of the proposed method?
2.	What is the relationship between the theoretical derivations and the proposed method?
3.	What is the motivation of using the VAE model?

---

> ### Author Response · Authors · 2023-11-22
> **Response to reviewer KTFt**
>
> We appreciate Reviewer KTFt's feedback and address the raised concerns as follows:
> 1. **Technical Novelty is Incremental**:
> We respectfully disagree with this assessment.Our paper is the first to apply a deep generative model for the generation of MILP instances with feasibility and boundedness guarantee. Please refer to the universal response for more details.
> 2. **Clarity on Theoretical Derivations and Motivation for Using VAE**:
> Our core contribution is to extract information from a well-designed generative space for producing new MILP examples with feasibility and boundedness guarantees. Theorem 1. in our paper provides theoretical backing for the guarantee, and the choice of generative model is not the key of our work.
> We acknowledge the reviewer's point about discussing our choice of generative model. Briefly, generating graphs from scratch, especially larger ones, might face complexity and scalability issues, hence our choice of VAE to modify parts of MILP examples iteratively. We will amend our paper to include comparisons with other models.
> 3. **Comparison with Recent Work at NIPS [5]**:
> Our work fundamentally differs from [5] and is concurrent, as detailed in our universal response.
> 4. **Adequacy of Experiments**:
> Thanks for your advice. We find most studies in the ML for MILP community focus on problems of a hundred variables to a thousand in size. Our experiments on two sizable MIPLIB datasets might sufficiently demonstrate the essence of our method.
> The downstream tasks suggested by the reviewer could they themselves be independent research branches and are far removed from the core issue we address.
> Furthermore, the framework of DIG-MILP supports general MILP generation, unlike G2SAT which works on stratification problems, making it unsuitable as a baseline in our context.
> We hope this adequately addresses the concerns and thank you for your valuable input.

---

> ### Comment · Reviewer_KTFt · 2023-11-22
> **Thanks for the authors' rebuttal**
>
> Thanks for the authors' response. However, my major concerns 3 and 5 have not been properly addressed.
>
> 1.	(Concern 3 in weaknesses) I suggested that the authors explain the motivation of using the VAE model rather than the other generative models in detail. However, the authors replied that the choice of generative model is not the key of their work, which did not directly address my concern.
> 2.	(Concern 5 in weaknesses) I suggested that the authors conduct three additional experiments, which are important for demonstrating the effectiveness of their proposed method. However, the authors did not provide these experiments.
>
> Given the current status of my communication with the authors, I would keep my score.

---

### Official Review · Reviewer_NJds · 2023-11-01

**Soundness:** 2 fair
**Presentation:** 2 fair
**Contribution:** 1 poor
**Rating:** 3
**Confidence:** 4

**Summary:**

This paper proposes a deep instance generator for MILPs with feasibility gurantee. It uses a VAE model trained on a dataset to generate similar MILP instances, and leverages a dual method proposed by Bowly et al. for feasibility.

**Strengths:**

1. This paper proposes a MILP generator with feasibility gurantee.
2. It conduct some experiments to demonstrate the effectiveness and for analysis.

**Weaknesses:**

1. The technical novelty is minor. The proposed model is a direct combination of two existing methods [1] and [2]. [1] is a recent work accepted by NeurIPS and this paper is almost the same with [1]. Even if taking [1] without consideration, this work is an application of existing techniques, i.e., the VAE for graph generation and the feasible instance construction method proposed in [2].
2. Does this paper deal with MILPs or IPs? In Eq. (1) all variables are constrained as integers. In table 1 there are no features indicating whether the variables are integers.
3. Do the datasets contain unfeasible MILPs? Can the model learn to generate feasible MILPs without the feasibility gurantee? The necessity of this component is not demonstrated with ablation study.
4. The proposed method does not performs better than random significantly.
5. Why not report the hyper-configuration results to show whether this method can benefit this task?
6. What is the useness of the optimal value prediction task? Can the proposed method help the solving instead of just predicting the optimal value?

[1] https://arxiv.org/abs/2310.02807

[2] Simon Andrew Bowly. Stress testing mixed integer programming solvers through new test instance generation methods. PhD thesis, School of Mathematical Sciences, Monash University, 2019.

**Questions:**

See weaknesses.

---

> ### Author Response · Authors · 2023-11-22
> **Response to reviewer Njds**
>
> Thank you, Reviewer NJds, for your detailed reading and suggestions on our paper. Below, we address the weaknesses raised by NJds:
> 1. **Technical Novelty is Minor**:
> We respectfully disagree with the reviewer’s assessment. Please refer to our universal response for clarification.
> 2. **MILPs vs. IPs Clarification**:
> Thank you for pointing this out. The short answer is yes, DIG-MILP supports generating MILP.
> It would only require training an additional classifier in the decoder to decide whether to round solution or slack variables into integers, which could then control whether the variables are discrete or continuous. When using DIG-MILP to solve IP instances, one could omit such classifiers. We will revise our manuscript to include such an explanation.
> 3. **Feasibility of Generated MILPs**:
> The short answer is the datasets do not contain infeasible MILPs. DIG-MILP indeed requires the training data with at least one set of solutions to generate feasible-bounded new instances. But we argue that such a requirement of training DIG-MILP solely on feasible datasets is both practical and appropriate. Please refer to universal response 3) for more detailed explanation.
> 4. **Performance Compared to Random**:
> We respectfully disagree with the reviewer’s argument. From the aspect of the structural similarity (15%-20% higher similarity score) and performance on the downstream tasks (10% higher Pearson correlation score in solver tuning and much lower MSE in ML model training), the performance of DIG-MILP is much better than the random baseline.
> 5. **Hyper-configuration results**:
> Hyper-configuration results can be inferred from the provided figures, where configurations resulting in shorter SCIP solving times on both original and generated data are preferred. Direct comparison of optimal hyper-configurations is challenging, because it may not be proper to evaluate the similarity of two sets of hyper-configurations by their distance directly on Euclidean space.
> 6. **Optimal Value Prediction Task**:
> Tackling the optimal value prediction task is a preliminary step in aiding MILP problems with ML models. We follow a general task setting introduced in [1]. Directly solving MILP is more challenging than predicting the optimal solution. In MILP generation task, one may first do well on the simpler task and then move towards more challenging tasks in the future work.
>
> We hope this addresses your concerns and thank you once again for your valuable feedback.
>
> [1] Chen et al. On the representation of solutions to elliptic pdes in barron spaces ICLR 2023.

---

### Author Response · Authors · 2023-11-22
**Universal Response to Reviewers**

We extend our gratitude to all reviewers for their meticulous reading of our paper and the valuable feedback provided. We would like to provide a universal response to In response to the commonly raised issues address these concerns. （E.g. Reviewer NJds, KTFt, T4Ma raised questions on the technique novelty of the paper, and asked the difference between DIG-MILP and [1]; reviewer NJds, HSpe and T4Ma raised questions on the motivation of utilizing only feasible-bounded datasets to generate feasible-bounded MILP instances only.)
1. **Article's Technical Novelty**:
   Our paper is the first to apply a deep generative model for the generation of MILP instances with feasibility and boundedness guarantee. By carefully designing the generative space structure. DIG-MILP is not a simple application of existing methods. Instead, DIG-MILP features specific designs tailored to MILP problems, targeting the structure of the VAE model, its input space (Table. 1), output space (Equation. 8), as well as the structure and mechanism of the generation space (Proposition 1). By this DIG-MILP ensures the feasibility and boundedness of the generated MILP instances as well as leads to promising results in two downstream tasks. To mention that, DIG-MILP is also the first that has been shown can generate new instances for solver hyper-configuration tuning without sharing the original data.

1. **Fundamental Differences from Paper [1]**:
*  DIG-MILP (ICLR2024 submission deadline on September 28th) is concurrent with Paper [1] (accepted by Neurips 2023 officially released on December 22nd, arXiv version released on Oct 4th) , whose code and paper are both online later than the ICLR paper submission deadline.

*  DIG-MILP is fundamentally different from [1]. The only similarity of methodology lies in the two works is the adoption of VAE as the generative model and the strategy of iteratively changing parts of the MILP instance. However, the key motivation, contribution and the problem to solve of DIG-MILP does not depend on the utilization of a specific generative model as mentioned above, Instead, DIG-MILP aims to generate feasible-bounded only MILP instances similar to the target dataset distribution with the aid of weak duality theory (that provides theoretical guarantee) and deep neural network (that extracts more in-depth feature than hand-craft methods).

3. **Focus on Generating Feasible Bounded Instances**:
   The emphasis on generating feasible bounded instances stems from the practical realities of MILP applications from two aspects.
*  First, generating only feasible-bounded MILP aligns with real-world scenarios. In real-world scenarios, most MILP problems modeled from practical problems are feasible, even though some problems could be infeasible after primary modeling, they are modified into feasible ones via various tricks before being fed into the solvers.
*  Second, the training data resource is mostly feasible-bounded only. Most of the available MILP datasets contains feasible-bounded MILP instance only, which makes it reasonable to train generative models on feasible-bounded MILP instances and generate feasible-bounded instances only.

[1] Geng, Zijie, et al. "A Deep Instance Generative Framework for MILP Solvers Under Limited Data Availability." Neural Information Processing Systems (2023).

---

### Meta-Review · Area_Chair_ap3w · 2023-12-05

**Metareview:**

This paper presents DIG-MILP, a deep generative framework based on VAE. All reviewers consistently believe the paper in its current form is not ready for publication. Reviewers are concerned about the novelty and the comparison of the work with prior papers, as well as other major drawbacks. I would suggest the authors revising their paper accordingly and prepare for a next version.

**Justification For Why Not Higher Score:**

All reviewers consistently believe the paper in its current form is not ready for publication, with a score of rejection (3).

**Justification For Why Not Lower Score:**

N/A

---

### Decision · Program_Chairs · 2024-01-16

Reject